# MICU1 drives glycolysis and chemoresistance in ovarian cancer

Prabir K. Chakraborty[1], Soumyajit Banerjee Mustafi[2], Xunhao Xiong[1], Shailendra Kumar Dhar Dwivedi[2], Vasyl Nesin[3], Sounik Saha[1], Min Zhang[4], Danny Dhanasekaran[3,5], Muralidharan Jayaraman[3], Robert Mannel[2,5], Kathleen Moore[2,5], Scott McMeekin[2,5], Da Yang[4], Rosemary Zuna[1,5], Kai Ding[6], Leonidas Tsiokas[3], Resham Bhattacharya[2,5] & Priyabrata Mukherjee[1,5]

Cancer cells actively promote aerobic glycolysis to sustain their metabolic requirements through mechanisms not always clear. Here, we demonstrate that the gatekeeper of mitochondrial $Ca^{2+}$ uptake, Mitochondrial Calcium Uptake 1 (MICU1/CBARA1) drives aerobic glycolysis in ovarian cancer. We show that MICU1 is overexpressed in a panel of ovarian cancer cell lines and that *MICU1* overexpression correlates with poor overall survival (OS). Silencing MICU1 *in vitro* increases oxygen consumption, decreases lactate production, inhibits clonal growth, migration and invasion of ovarian cancer cells, whereas silencing *in vivo* inhibits tumour growth, increases cisplatin efficacy and OS. Mechanistically, silencing MICU1 activates pyruvate dehydrogenase (PDH) by stimulating the PDPhosphatase-phosphoPDH-PDH axis. Forced-expression of MICU1 in normal cells phenocopies the metabolic aberrations of malignant cells. Consistent with the *in vitro* and *in vivo* findings we observe a significant correlation between MICU1 and pPDH (inactive form of PDH) expression with poor prognosis. Thus, MICU1 could serve as an important therapeutic target to normalize metabolic aberrations responsible for poor prognosis in ovarian cancer.

[1] Department of Pathology, The University of Oklahoma Health Sciences Center, Oklahoma City, Oklahoma 73104, USA. [2] Department of Obstetrics and Gynecology, The University of Oklahoma Health Sciences Center, Oklahoma City, Oklahoma 73104, USA. [3] Department of Cell Biology, The University of Oklahoma Health Sciences Center, Oklahoma 73104, USA. [4] Department of Pharmaceutical Sciences, University of Pittsburgh School of Pharmacy, Pittsburgh, Pennsylvania 15261, USA. [5] Peggy and Charles Stephenson Cancer Center, The University of Oklahoma Health Sciences Center, Oklahoma City, Oklahoma 73104, USA. [6] College of Public Health, The University of Oklahoma Health Sciences Center, Oklahoma City, Oklahoma 73104, USA. Correspondence and requests for materials should be addressed to P.M. (email: Priyabrata-Mukherjee@ouhsc.edu).

Mounting evidence indicates that deranged metabolism, particularly aerobic glycolysis, is linked to tumour growth and chemoresistance[1–3]. First described by Otto Warburg in 1930s (ref. 4), aerobic glycolysis is now recognized to be a major metabolic requirement for tumours to grow and resist therapy. Many enzymes in the glycolytic pathway are emerging targets in anticancer therapy and, in combination with chemotherapy, are showing promising results[5]. Several enzymes in dysregulated fatty acid and glutamine metabolism have also been linked to tumour growth and chemoresistance[6]. However, key molecular machinery that regulates the metabolic demand between mitochondrial pyruvate oxidation and glycolysis is still elusive.

A key rate-limiting step that determines the metabolic fate between glycolysis versus mitochondrial oxidative phosphorylation is the conversion of pyruvate to acetyl CoA by pyruvate dehydrogenase (PDH) (ref. 7). Consequently, pyruvate dehydrogenase kinase (PDK) that phosphorylates PDH to its inactive phosphorylated-PDH (pPDH) form has been shown to promote glycolysis[4]. Hence, the disruption of PDK-PDH axis could decimate cancer progression and chemoresistance. In addition to pathogenic mutations or depletion of the mitochondrial genome, mitochondrial $Ca^{2+}$ homeostasis can contribute to development of chemoresistance in malignant tumours[8]. Although alterations in $Ca^{2+}$ signalling may not be a requirement for the initiation of cancer, the consequences of altered $Ca^{2+}$ transport in cancer cells may contribute to tumour progression and drug resistance[9]. Characterizing such changes may help to identify new therapeutic targets. Indeed, the main plasma membrane-bound $Ca^{2+}$ transporters that may be involved in the development of multi-drug resistance (MDR) include store-operated channels (SOC), transient receptor potential channels (TRPs), voltage-gated $Ca^{2+}$ channels and plasma membrane $Ca^{2+}$ ATPases[10]. SOCs are activated through a mechanism in which depletion of intracellular $Ca^{2+}$ stores leads to aggregation of Stromal interaction molecule 1 (STIM1), that is, the $Ca^{2+}$ sensor in endoplasmic reticulum (ER), and Orai1, the membrane-bound $Ca^{2+}$ channel protein[11]. Reduced expression of Orai1, and, consequently, reduced SOC activity, prevents $Ca^{2+}$ overload in response to pro-apoptotic stimuli and thus establishes the MDR phenotype in prostate cancer cells[9]. On the other hand, Faouzi et al.[12] suggest that Orai3 promotes apoptosis resistance in breast cancer cells. Several of the TRP channels have been implicated in the development of MDR, for example, TRPC1, TRPV2 and TRPV6 (ref. 9). However, the key molecular machinery that unifies divergence in $Ca^{2+}$ signalling to glycolysis and chemoresistance has not been identified.

Recently, mitochondrial $Ca^{2+}$ 'uniporter' (MCU), a calcium-selective ion channel responsible for low-affinity $Ca^{2+}$ uptake into the mitochondrial matrix has been identified[13,14]. MCU influences a number of cellular $Ca^{2+}$-dependent processes, including cell death[15]. The function of MCU is regulated by another recently identified protein, mitochondrial $Ca^{2+}$ uptake 1 (MICU1) that resides in the inner mitochondrial membrane[16]. While MCU expression is correlated with cancer progression in sporadic studies (prostate, colon and breast), the functional role for MICU1 in cancer is unknown[17]. Furthermore, recent reports suggest that in the absence of MICU1, the mitochondria become constitutively loaded with $Ca^{2+}$, resulting in excessive ROS generation and sensitivity to apoptotic stress[15]. In this context, our previous work has shown that small interfering RNA-mediated silencing of MICU1 sensitized OvCa cells to positively charged gold nanoparticles[18], suggesting for the first time a potential role of MICU1 in drug resistance.

Here, we reveal a unique function of MICU1 that regulates metabolic fate and confers chemoresistance in OvCa. These findings provide molecular insight into abnormal metabolism in OvCa linked to MICU1 and its role in poor prognosis. Our results purport possible therapeutic application of MICU1 targeting as an innovative way to normalize aberrant metabolism in cancer to improve poor prognosis.

## Results

**Expression and pathological significance of MICU1 in cancer.** Three different MICU isoforms exist in vertebrates, while only two are conserved in protozoa and plants. Tissue distribution studies in mice have revealed MICU1 to be broadly expressed and detected in 12 out of 14 mouse tissues, with the skeletal muscle and kidney having the highest expression of MICU1 mRNA (ref. 19). Moreover, whereas MICU1 and MICU2 have broad tissue expression, MICU3 appears to be expressed at significant levels only in the nervous system[20]. To understand the role of MICU1 in human cancers we sought to determine the expression of MICU1 in normal versus ovarian cancer (OvCa) cell lines by immunoblotting. The origin of OvCa has been debated and suggested to be originating from the ovarian fallopian tube epithelium instead of the ovarian surface epithelium[21]. Recently Domcke et al. have characterized OvCa cell lines that more closely resemble cognate tumour profiles and that can be a true representative of high grade serous ovarian cancer (HGSOC) (ref. 22). We pursued our study bearing this current understanding for cell line utilization. Interestingly, there was minimal to no expression of MICU1 in the non-malignant ovarian surface epithelial cell line (OSE) or fallopian tube epithelium derived cell line FTE188, while the OvCa cell lines had significant expression for MICU1 (Fig. 1a). To understand the role of MICU1 in chemoresistant and aggressive OvCa cell line models, we selected CP20 and OV90 cell lines for our subsequent studies. The CP20 cell line was developed by sequential exposure of an OvCa cell line to increasing concentrations of cisplatin and presents a chemoresistance cell line model[23]. The OV90 cell line was derived from ascites of a grade 3, stage IIIC OvCa patient and hence provides a clinically relevant aggressive model to study roles of MICU1 in vitro and in vivo[24]. Next we investigated the clinical relevance of MICU1 expression in OvCa. We used clinically annotated mRNA data from GEO (Gene Expression Omnibus) databases[25]. The mRNA expression data for all genes were detected by microarray. The clinico-pathologic and platform information of these cohorts are provided in supporting information (Supplementary Table 1). Quantitative analysis for Kaplan-Meier overall survival curves in MICU1 low and high expression OvCa cases in GSE32062 and GSE26712 cohorts showed significant correlation between high MICU1 expression and poor overall survival (Fig. 1b). To determine a role of MICU1 in chemoresistance, we assessed the expression of MICU1 protein in an OvCa tissue microarray (TMA) (Fig. 1c) consisting of 126 drug-resistant HGSOC patient tissues by immunohistochemistry analysis. Among 126 tumour tissues, almost three-quarters ($n = 92$, 73%) of samples had a moderate to high expression of MICU1 with a 95% Clopper-Pears on exact confidence interval (64%, 81%). Since MICU1 is overexpressed both in OvCa cell lines and primary ovarian cancers and overexpression correlated with poor survival, we next sought to determine the functional significance of MICU1 in OvCa growth and drug resistance.

**Role of MICU1 in regulating cancer phenotype.** It is reported that MICU1 regulates mitochondrial $Ca^{2+}$ uptake[26] and $Ca^{2+}$ homeostasis plays critical roles in numerous cancer phenotypes[27]. Therefore, to establish a role of MICU1 in OvCa cell phenotypes, we utilized siRNA or lentivirus mediated shRNA to silence

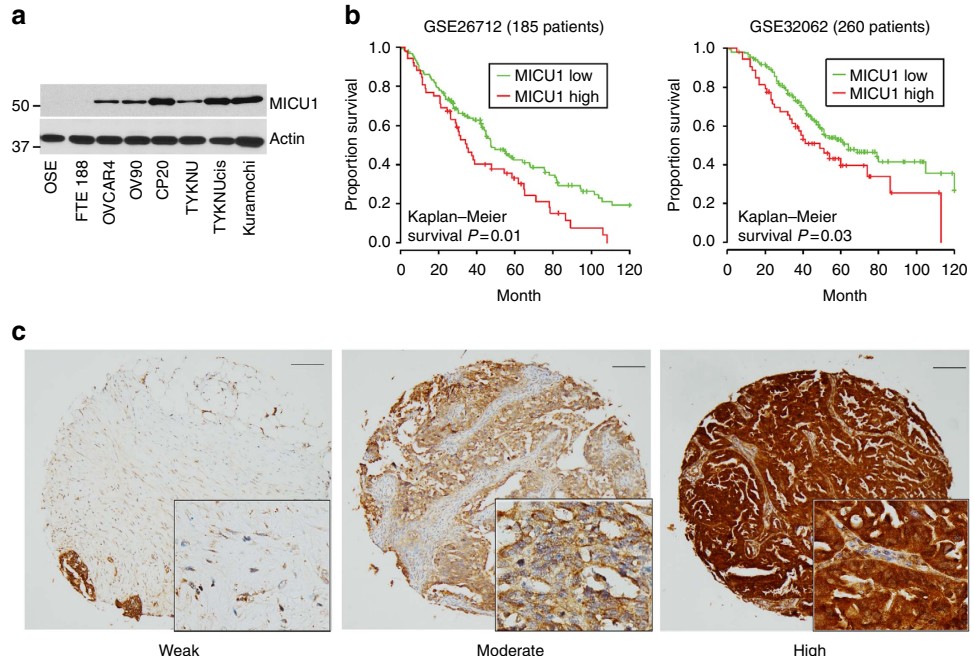

**Figure 1 | Expression and pathological significance of MICU1 in OvCa.** (**a**) Expression of MICU1 in various ovarian cell lines as determined by immunoblotting. Actin is used as the loading control. (**b**) Kaplan-Meier overall survival curves in MICU1 low and high expression OvCa cases from GEO databases. The proportion survival is plotted versus time since diagnosis in months. Kaplan-Meier curves with a log-rank test where $P$ value $< 0.05$ considered significant. (**c**) Immunohistochemical (IHC) staining of a tissue microarray of drug-resistant epithelial OvCa samples for MICU1 (1:150). Representative images taken at $\times 4$ magnification are shown of (i) weak, (ii) moderate and (iii) high staining. Inset shows magnified areas of individual IHC stains taken at $\times 20$ magnification. Scale: 100 μm.

MICU1. We generated stable clones expressing shMICU1 (that target different MICU1 mRNA sequences) in OV90 cells (Supplementary Fig. 1) and selected clone 3 (C3) for all subsequent experiments and henceforth depicted as shMICU1-OV90. Notably, the silencing of MICU1 did not interfere with the mRNA expression of other MCU complex, namely MCU and essential MCU regulator[20], while mitochondrial calcium uptake 2 (MICU2) showed modest decrement (Supplementary Fig. 2), in agreement with previous reports[28]. Moreover, mitochondrial copy number remains unaltered upon silencing of MICU1 in both CP20 and OV90 cells (Supplementary Fig. 3). The role of $Ca^{2+}$ on clonal growth and differentiation has been well established in human bronchial, leukemic and epidermal cells[29]. The anchorage independent clonal growth of cancer cells in semi-solid medium reflects the potency of tumour cells to survive and grow in secondary locations *in vivo* and correlates closely with tumorigenicity in animal models[30].

Since MICU1 is highly expressed in chemoresistant HGSOC tissues and clonal growth is implicated in aggressive phenotype including drug resistance[31,32], we wanted first to investigate a role of MICU1 in clonal growth. We evaluated the effect of MICU1 silencing on anchorage independent clonal growth in CP20 and OV90 cells. Compared to the control, significant reduction in number of colonies in siMICU1 (decreased by ∼81% in CP20 and ∼82% in OV90) or shMICU1-OV90 (decreased by ∼76%) was observed (Fig. 2a). Inhibition of clonal growth upon MICU1 silencing implicates a role of MICU1 for OvCa growth and metastasis that involves cell migration and invasion. Indeed $Ca^{2+}$ homeostasis affects cellular migration and invasion and numerous $Ca^{2+}$ channels have been reported to be involved in cancer cell migration, invasion or both[11,33]. To identify and provide evidence for a role of MICU1 in cell migration and invasion, we performed migration and invasion studies after transiently silencing OvCa cell lines with siRNA of MICU1 and

comparing them with scrambled controls. OV90 cells transfected with scrambled control (siCTL-OV90) migrated efficiently towards an FBS gradient (Fig. 2b, upper left panel), whereas silencing of endogenous MICU1 expression resulted in a marked decrease in the cell migration (Fig. 2b, lower left panel). Similar reductions in cell migration phenotype were obtained in siMICU1-CP20 cells and shMICU1-OV90 cells. Quantification of results indicated that the silencing of MICU1 attenuated CP20 cell migration by 84%, OV90 cells by 80% and shMICU1-OV90 cells by 69%, respectively (Fig. 2b, right panel). MICU1 silencing, however, showed no significant decrease in cell proliferation during the course of the migration study, confirming that the decrease in cell migration upon MICU1 silencing is due to the effect on cell migratory pathways and not due to a decrease in cellular proliferation. We also examined whether MICU1 affects the cellular invasion in OvCa using several complimentary approaches involving Boyden chamber and gelatin matrix degradation assay. A significant decrease in invasion of siMICU1-OV90 (Fig. 2c, left panel), or siMICU1-CP20 and shMICU1-OV90 cells was observed when compared to their respective controls. Quantification of invading cell numbers indicated that downregulation of MICU1 expression attained both transiently (siMICU1) or stably (shMICU1) reduced invasion of siMICU1-CP20 by 74%, siMICU1-OV90 cells by 68% and shMICU1-OV90 cells by 69% (Fig. 2c, right panel). To further strengthen our observations we performed cell migration (Fig. 2d) and cell invasion (Fig. 2e) studies utilizing the fluorescence dye Calcein-AM. Transient silencing of MICU1 or stable knockdown of MICU1 caused significant impairment in cellular motility. In an Oregon Green 488 Gelatin matrix the cell invasion was examined (Fig. 2f) which revealed MICU1 silencing reduced the matrix degrading ability of CP20 cells by ∼77% and that of OV90 cells by ∼53%, suggestive of lowered cellular invasion (Fig. 2g). Thus, silencing of MICU1 reduces anchorage-

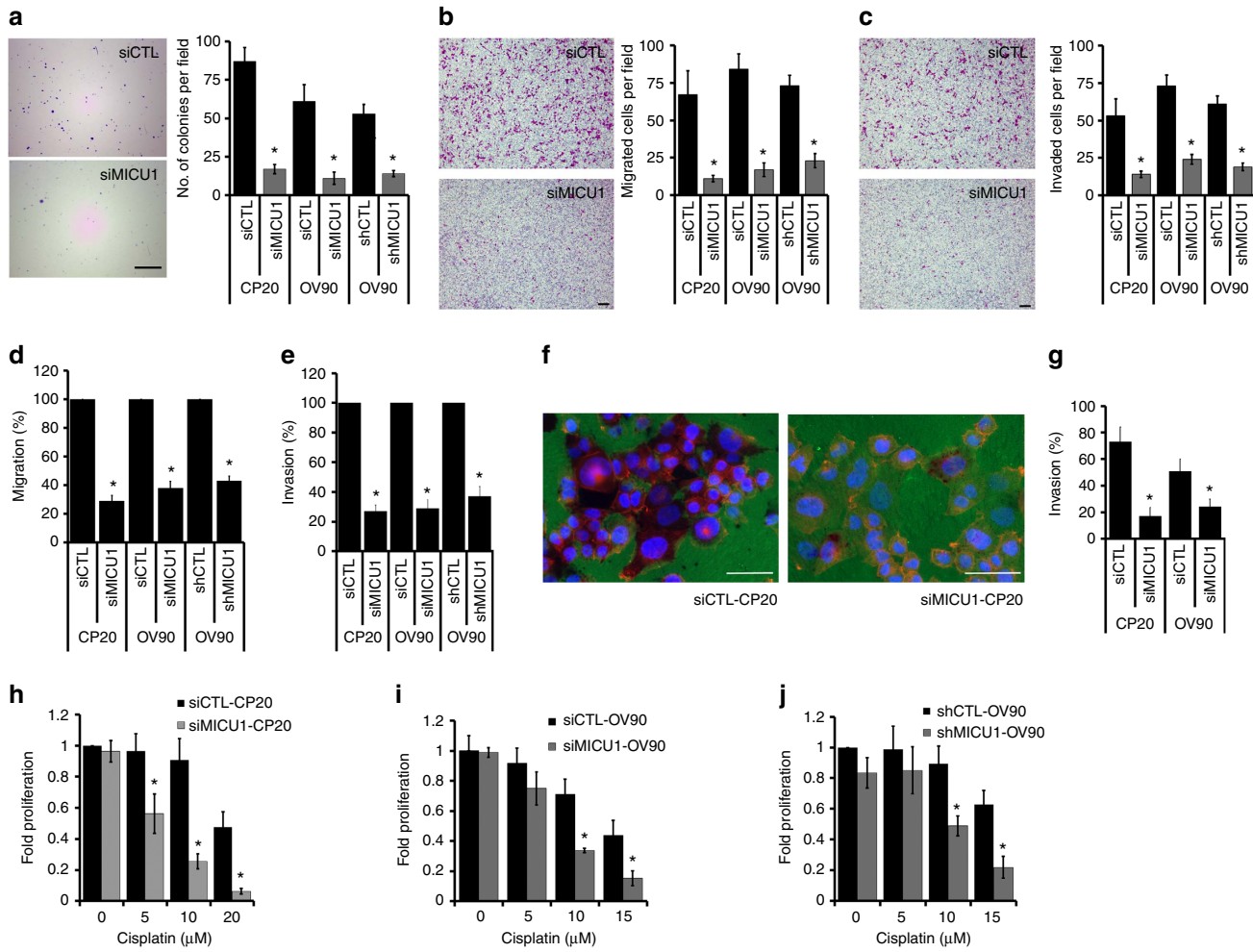

**Figure 2 | Effect of MICU1 on cancer phenotypes.** (**a**) MICU1 silencing affects clonal growth of OvCa cells. CP20 and OV90 cells were transfected with MICU1 siRNA or control siRNA and 48 h after transfection cells were re-plated in 0.3% agar. Similarly, stable cell lines shCTL-OV90 and shMICU1- OV90 were also plated in 0.3% agar. After 10 (CP20) and 14 (OV90) days, colonies were stained and quantified. Values are represented as mean ± s.d. (**b**) Silencing of MICU1 inhibits cell migration in OvCa cells (CP20 and OV90). Migration of siCTL and siMICU1 or shCTL and shMICU1 cells towards serum gradient was examined and number of cells per field was counted. Values are represented as mean ± s.d. (**c**) Silencing of MICU1 inhibits cell invasion of fibronectin matrix by OvCa cells (CP20 and OV90). Invasion of siCTL and siMICU1 or shCTL and shMICU1 cells through fibronectin-coated filters was examined using Boyden chamber and number of cells per field was counted and values are represented as mean ± s.d. (**d**) Cell migration assay performed in Boyden chamber followed by Calcein-AM staining and fluorescence measurement. (**e**) Cell invasion assay performed in Boyden chamber coated with fibronectin followed by Calcein-AM staining and fluorescence measurement. (**f**) Representative images of siCTL or siMICU1 transfected CP20 cells plated on Oregon Green 488 Gelatin coated coverslips for 48 h, fixed and stained with Alexa Fluor 555 Phalloidin. (**g**) Quantification of CP20 or OV90 cells degrading the gelatin matrix and values are mean ± s.d. (**h,i**) CP20 (**h**) and OV90 (**i**) cells were transfected with siCTL or siMICU1 for 24 h and cells were re-plated in 96-well plates. Cells were treated with cisplatin at the indicated concentrations and proliferation was determined using CyQUANT NF cell proliferation assay after 48 h. Mean fold change ± s.d. in cell proliferation was evaluated by comparing with respective cells receiving similar treatment. (**j**) shCTL-OV90 or shMICU1-OV90 cells were plated in 96-well plates. Cells were treated with cisplatin at the indicated concentrations and proliferation was determined as in **i**. *$P<0.05$ was statistically significant when compared to siCTL/shCTL (**a–g**) or similarly treated siCTL/shCTL (**h–j**). Two-sided Student's $t$-test with unequal variances was used for statistical analysis. All the experiments were repeated independently at least three times and in triplicate. Scale bar (**a**) 2 mm and (**b,c** and **f**) 20 μm.

independent clonal growth and cellular mobility in OvCa cells, two pivotal hallmarks of cancer progression[34].

In our previous work, we demonstrated that MICU1 protected OvCa cells from the cytotoxic effects of positively charged gold nanoparticles ( +AuNPs) (ref. 18). Recently, Mallilankaraman et al.[35] showed that ceramide-induced HeLa cell death was enhanced by nearly 100% in MICU1 knockdown cells. These reports provide evidence of a potential role of MICU1 in drug resistance in OvCa and inspire us to investigate whether silencing MICU1 could sensitize OvCa cells to FDA approved drugs such as cisplatin. Cisplatin treatment for 48 h post MICU1 silencing

shows drug sensitization and significant effects on cellular proliferation in OV90 and CP20 cells over a range of cisplatin concentrations (5-20 μM) as shown in (Fig. 2h–j). While 20 μM cisplatin reduced the proliferation of MICU1 silenced CP20 cells by ∼94% compared to 53% in the control cells, 15 μM cisplatin reduced the proliferation of MICU1 silenced OV90 cells by ∼85% compared to ∼57% in control cells (Fig. 2h–j). We additionally employed BrDU incorporation assay to monitor the effects of cisplatin exposure on the cell proliferation in siCTL or siMICU1 OvCa cells CP20 and OV90 (Supplementary Fig. 4). Treatment with cisplatin significantly reduced BrDU

incorporation in MICU1-silenced CP20 and OV90 cells as compared to scrambled siRNA control, indicating enhanced effect of cisplatin to inhibit proliferation of OvCa cells upon MICU1 silencing (Supplementary Fig. 4). These results indicate a potential role of MICU1 in chemosensitization in epithelial ovarian cancers. We further attempted to test whether silencing of MICU1 sensitizes OvCa cells to cisplatin exclusively or similar effects prevail with other chemotherapeutic drugs.

We tested three additional commonly used chemotherapeutic drugs in OvCa therapy (topotecan, paclitaxel and doxorubicin) and MICU1 silencing showed significant sensitization of CP20 cells towards the aforesaid agents (Supplementary Fig. 5). The doses of these drugs were selected based on published reports[36]. Summarily, topotecan (10 μM) reduced the proliferation of MICU1 silenced CP20 cells by ~89% compared to ~62% in the control cells, paclitaxel (25 nM) reduced the proliferation of MICU1 silenced CP20 cells by ~85% compared to 57% in the control cells and doxorubicin (2 μM) reduced the proliferation of MICU1 silenced CP20 cells by ~88% compared to 68% in the control cells. These observations suggest that MICU1 provides survival advantage to cancer cells against cytotoxic insult and imparts drug resistance phenotype. We further attempted to explore any possible roles of MICU2 in OvCa chemoresistance as MICU2 is a key component of MCU complex and is involved in $[Ca^{2+}]_m$ homeostasis[37]. The expression of MICU2 was moderately higher in most of the OvCa cells as compared to non-malignant (OSE or FTE-188) cells (Supplementary Fig. 6A). Silencing of MICU2 was achieved by transient transfection with siRNA (Supplementary Fig. 6B). Interestingly, silencing of MICU2 did not affect the expression of MICU1 Supplementary Fig. 6B), but silencing of MICU1 caused modest decrease in MICU2 expression (Supplementary Fig. 6C), in agreement with studies reported by Patron et al.[38]. We next determined the effects of MICU2 silencing towards cisplatin sensitization in CP20 and OV90 cells. Interestingly, no enhanced activity of cisplatin was observed either in CP20 or OV90 cells upon MICU2 silencing. In MICU2-silenced CP20 cells treatment with 20 μM cisplatin reduced the proliferation by ~53% as compared to ~48% in scrambled control (Supplementary Fig. 6D). Similarly in MICU2-silenced OV90 cells treatment with 15 μM cisplatin reduced the proliferation by ~56% as compared to ~62% in siRNA scrambled controls cells (Supplementary Fig. 6E). These results further support our hypothesis that MICU1 and not MICU2 is a key player inducing chemoresistance in OvCa.

**Role of MICU1 in regulating apoptosis**. Given its recent identification, the functions of MICU1 beyond its mitochondrial $Ca^{2+}$ uptake are currently unknown[39]. Mitochondrial membrane depolarization is a pre-requisite for apoptosis by facilitating the release of cytochrome c and other apoptotic inducers[40]. In CP20 and OV90 cells loaded with the mitochondrial membrane potential indicator TMRE (tetramethylrhodamine, ethyl ester), we found that cisplatin produces modest membrane depolarization, while MICU1 silencing using siRNA substantially enhanced cisplatin induced membrane depolarization. Mitochondrial membrane potential ($\Delta\Psi_m$) is the driving force behind the influx of ions ($H^+$, $Ca^{2+}$) into mitochondria. As shown in Fig. 3a,b, MICU1 silencing caused significant drop in TMRE retention upon cisplatin treatment, even at low concentrations (5 μM), in CP20 and OV90 cells suggesting targeting MICU1 potentiates cisplatin action on OvCa cells. The contribution of MICU1 to enhance cellular apoptosis was additionally tested using western blot analysis for apoptosis markers such as cleaved caspase -3, -9 and -PARP in OV90 cells after cisplatin exposure (Fig. 3c). We observed in MICU1 silenced

cells that 10 μM cisplatin could selectively cleave caspases -3 and -9 as well as PARP. These aforesaid proteins are established markers of apoptosis induction, suggesting MICU1 silencing sensitizes cells towards cisplatin-induced apoptosis. However, how MICU1 silencing regulates the apoptotic machinery is not the focus of the current investigation. Quantification of apoptotic cell numbers revealed that CP20 or OV90 cells transiently transfected with siMICU1, upon cisplatin (10 μM) treatment showed ~1.93 fold and ~2.3 fold apoptosis compared to their respective controls (Fig. 3d). Overall, these results support the idea that MICU1 protects malignant cells against cisplatin-induced cytotoxicity. Excessive $[Ca^{2+}]_m$ overload can have deleterious effects, including inner membrane depolarization, mtROS (mitochondrial reactive oxygen species) overproduction, sensitization to apoptotic and necrotic stimuli, and activation of the permeability transition pore and subsequent cell death pathways[40]. While low mtROS levels are required to maintain normal cellular functions, aberrant mtROS production leads to oxidative stress leading to cellular damage and stress responses in cancer cells[41]. Indeed mtROS mediated mitochondrial dysfunctions in Jurkat cells potentiates TRAIL-induced apoptosis[42]. In our system, MICU1 silencing in CP20 cells resulted in strong staining for MitoSOX dye, an indicator for mtROS (Fig. 3e), suggesting MICU1 silencing generates oxidative stress in the mitochondria and causes vulnerability towards cytotoxic insults. To assess the importance of mtROS in inducing apoptosis, we treated cells with mtROS scavenger mitoTEMPO which can efficiently scavenge the mtROS (Supplementary Fig. 7). The apoptotic cell number in cisplatin treated siMICU1-CP20 cells were significantly reduced from ~2.2 fold to ~1.2 fold in the presence of mitoTEMPO (Fig. 3f). These results suggest that downregulating MICU1 expression generates mtROS in OvCa cells allowing cisplatin to induce a robust mitochondrial membrane depolarization. These differences execute downstream effects of sensitizing malignant cells to the anti-proliferative and/or pro-apoptotic effects of cisplatin. To further test the protective role of MICU1 against cisplatin toxicity we ectopically overexpressed MICU1 in normal OSE cells (Supplementary Fig. 8) followed by cisplatin treatment. Overexpression of MICU1 imparts resistance to cisplatin-induced inhibition of cell proliferation. Control OSE cells expressing empty vector (EV) showed a significant ~73% inhibition of cell proliferation compared to ~52% inhibition in MICU1 expressing cells (Supplementary Fig. 9). This result indicates that MICU1 imparts resistance towards chemotherapeutic drugs.

**Role of MICU1 regulating $[Ca^{2+}]_m$ in response to cisplatin**. To determine whether the mitochondrial $Ca^{2+}$ uniporter contributes to cellular responses to cisplatin by regulating $Ca^{2+}$ overload in mitochondria of OvCa cells, we performed Rhod-2AM fluorescence based mitochondrial $Ca^{2+}$ studies[18]. The mitochondria of MICU1-silenced cells have elevated basal $[Ca^{2+}]_m$ as compared to the wild-type cells (Fig. 4a,b). These results suggest that in the absence of MICU1, MCU-mediated $Ca^{2+}$ uptake is constitutively active in unstimulated cells. Most chemotherapeutic drugs cause a rise in $[Ca^{2+}]_m$ (ref. 43). Indeed, treatment with cisplatin (10 μM) increased $[Ca^{2+}]_m$ in MICU1-silenced CP20 (siMICU1-CP20) cells as compared to scrambled control (siCTL-CP20). The $[Ca^{2+}]_m$ spiked within 10 μM of the cisplatin treatment (Fig. 4a). However, cisplatin treatment in siCTL-CP20 or siCTL-OV90 cells did not result in a significant increase in $[Ca^{2+}]_m$ (Fig. 4b,c). We then utilized a fluorescent protein-based mitochondrial $Ca^{2+}$ indicator, GCaMP2-mt (ref. 44) to further confirm a role of MICU1 in protecting

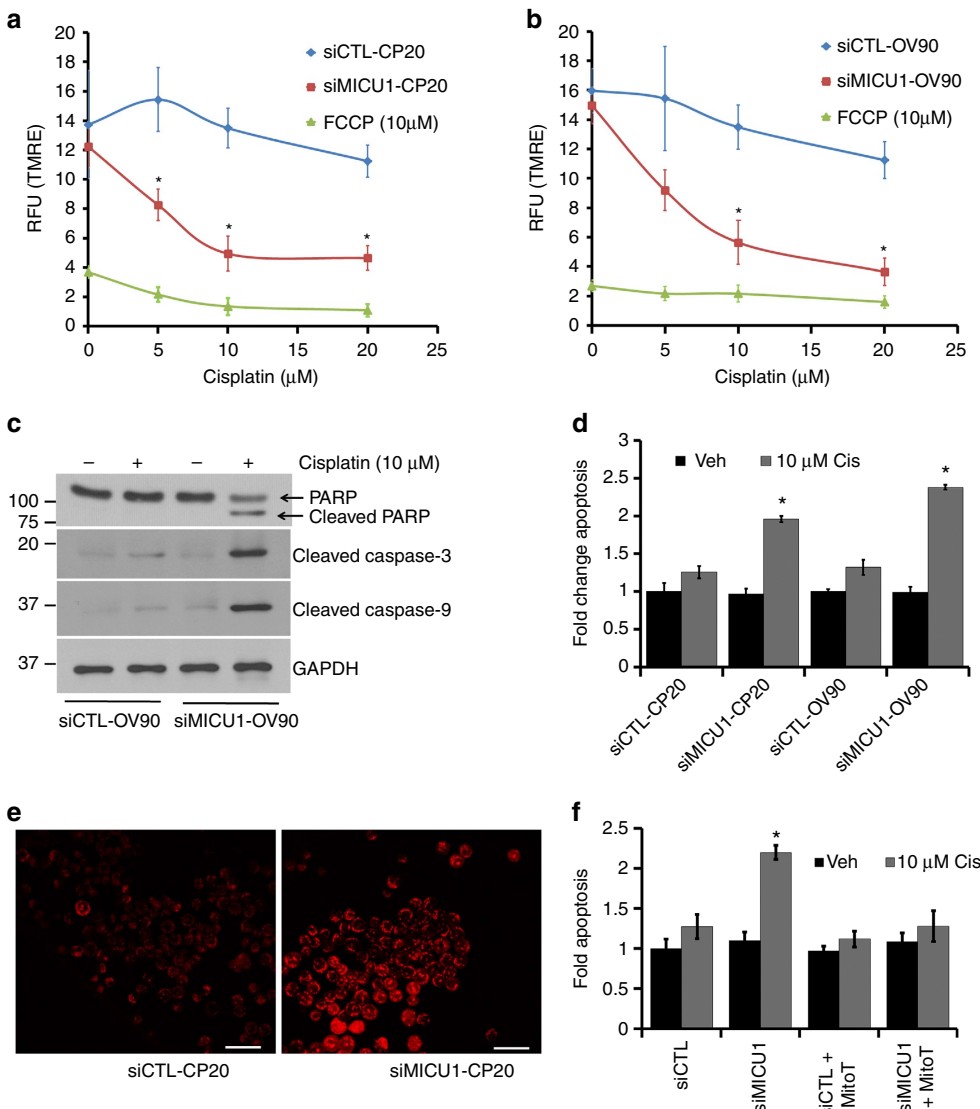

**Figure 3 | MICU1 modulates $\Delta\Psi_m$ responses and apoptosis in OvCa cells.** (**a,b**) In siCTL-/siMICU1-CP20 cells (**a**) and siCTL-/siMICU1-OV90 cells (**b**), loaded with the fluorescent mitochondrial membrane potential ($\Delta\Psi_m$) indicator TMRE, cisplatin caused significant membrane depolarization in MICU1 silenced cells compared to respective siCTL cells receiving similar treatment values are represented as mean ± s.d. FCCP was used as a negative control. (**c**) siCTL/siMICU1-OV90 treated or untreated with cisplatin were immuno-probed for apoptotic markers (cleaved PARP, cleaved caspase-3 and cleaved caspase-9). GAPDH was used as a loading control. (**d**) Quantification of apoptotic cell number in CP20 and OV90 cells upon cisplatin treatment (10 μM). Silencing MICU1 sensitized cells to cisplatin-induced apoptosis and values are mean ± s.d. (**e**) siCTL-/siMICU1-CP20 cells grown on coverslips were incubated with mtROS detecting agent MitoSOX (5 μM) and then imaged. (**f**) mtROS contributes to cisplatin sensitization in MICU1 silenced cells. Cisplatin (10 μM) treatment in CP20 cells in presence of mtROS scavenger MitoTempo (10 μM) failed significantly to induce apoptosis in siCTL- or siMICU1-CP20 cells and values are represented as mean ± s.d. *$P < 0.05$ when compared to siCTL (**a,b**) or vehicle (**d,f**). Two-sided Student's $t$-test was used for statistical analysis with unequal variances. Statistical significance was set at $P < 0.05$. All the experiments were repeated independently at least three times and in triplicate.

mitochondria from cisplatin induced $Ca^{2+}$ overload and to complement Rhod-2 AM data. Mitochondria specific GCaMP2-mt was transiently expressed in shCTL-OV90 and shMICU1-OV90 cells and changes in fluorescence in response to cisplatin was measured by single cell $Ca^{2+}$ imaging (Fig. 4d). Depletion of MICU1 led to an approximately ~4-fold increase in maximum response (Fig. 4d). Enhanced fluorescence was also observed in cisplatin-treated siMICU1-CP20 cells as compared to cisplatin-treated siCTL-CP20 cells, but to a lesser degree (Supplementary Fig. 10). To further confirm the protective role of MICU1 against cisplatin driven rise in $[Ca^{2+}]_m$, we ectopically overexpressed MICU1 in normal OSE cells (Supplementary Fig. 8) followed by cisplatin treatment and measured $[Ca^{2+}]_m$ from GCaMP2-mt

fluorescence. While cisplatin treatment increased $[Ca^{2+}]_m$ in OSE cells, however, such $[Ca^{2+}]_m$ overload was prevented in ectopically expressed MICU1-OSE cells (Fig. 4e). We further confirmed our study in another nonmalignant FTE-188 cell after ectopic overexpression of MICU1. The results demonstrate that ectopic expression of MICU1 in FTE-188 cells reduced GCaMP2-mt fluorescence approximately ~6-fold (in maximum response) as compared to wild-type FTE-188 cells (Fig. 4f). Since MICU1 and MICU2 has been reported to play opposing roles in MCU mediated $[Ca^{2+}]_m$ uptake[38], we wanted to investigate whether MICU2 had any effect on cisplatin-induced increase in $[Ca^{2+}]_m$. Utilizing GCaMP2-mt probe, we determined changes in $[Ca^{2+}]_m$ in response to cisplatin in MICU2 silenced shCTL-OV90 or

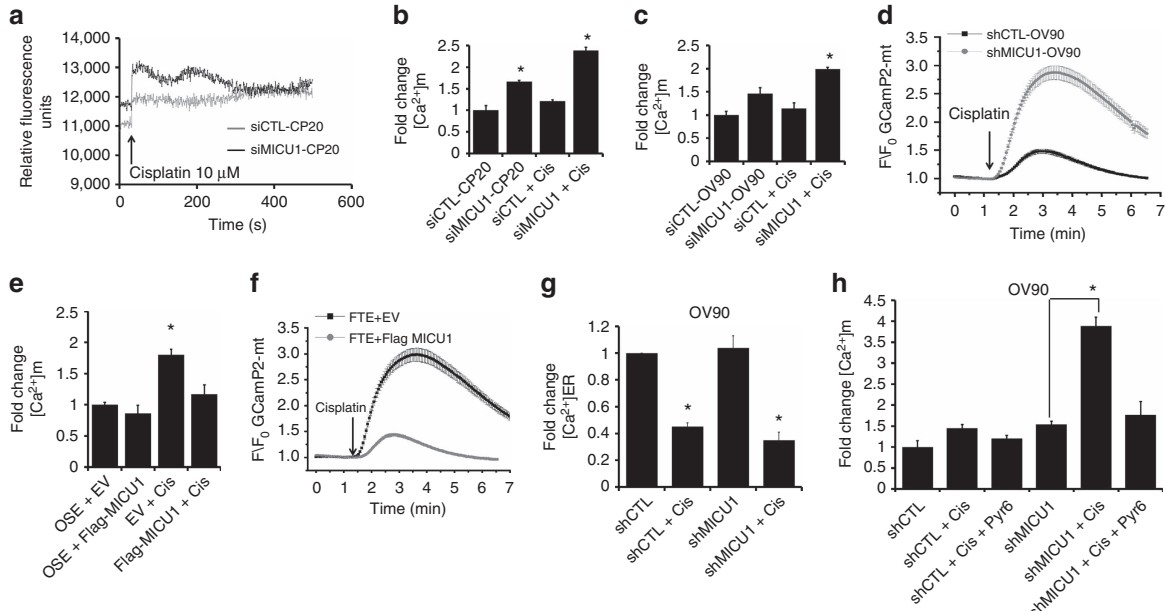

**Figure 4 | MICU1-mediated $Ca^{2+}$ buffering in OvCa cells evaluated using cisplatin. (a)** CP20 cells loaded with the fluorescent dyes Rhod-2AM (5 μM), 10 μM cisplatin caused rapid increase in $[Ca^{2+}]_m$. **(b,c)** Silencing of MICU1 by siRNA increased $[Ca^{2+}]_m$ responses to cisplatin in CP20 (B) or OV90 (c) cells loaded with Rhod-2AM (5 μM) and values are mean ± s.d. **(d)** Normalized GCaMP2-mt fluorescence in shRNA transfected (shCTL-OV90) or MICU1-specific shRNA transfected (shMICU1-OV90) cells stimulated with cisplatin (10 μM). The experiment was repeated three times (total no. of cells cell quantified: shCTL-OV90 $n = 158$ and shMICU1-OV90 $n = 128$). **(e)** Changes in $[Ca^{2+}]_m$ in OSE cells expressing Flag-MICU1 upon 10 μM cisplatin treatment as compared to empty vector (EV) transfected cells and values are mean ± s.d. **(f)** Normalized GCaMP2-mt fluorescence in FTE188 cells transfected with empty vector (FTE + EV) or Flag-MICU1 (FTE + Flag-MICU1) cells stimulated with cisplatin (10 μM). The experiment was repeated three times (total no. of cells quantified: FTE + EV $n = 135$ and FTE + Flag MICU1 $n = 433$). **(g)** Cisplatin causes similar release of $[Ca^{2+}]_{ER}$ in presence or absence of MICU1 as determined in OV90 cells loaded with $[Ca^{2+}]_{ER}$ indicator dye Fluo-5N (5 μM) and values are mean ± s.d. **(h)** Disruption of $[Ca^{2+}]_{ER}$ release by Pyr6 (50 nM) prevents cisplatin-induced changes in $[Ca^{2+}]_m$ of OV90 cells. Cells were pretreated with Pyr6 (50 nM) and Rhod-2AM (5 μM) for 15 min followed by 10 μM cisplatin treatment, fluorescence for Rhod-2AM was determined. Data are expressed as mean ± s.d. *$P < 0.05$ for comparisons among more than two groups, we performed ANOVA with Bonferroni's correction. $P < 0.05$ was considered statistically significant. All tests were two-sided. Horizontal line represents statistical comparison between respective groups. All the experiments were repeated independently at least three times and in triplicate.

shMICU1-OV90 cells (Supplementary Fig. 11). Silencing MICU2 resulted in lowering of $[Ca^{2+}]_m$ upon cisplatin stimulation in both the cells (Supplementary Fig. 11), suggesting opposing roles of MICU1 and MICU2 towards cisplatin induced changes in $[Ca^{2+}]_m$ in OvCa cells.

Next we wanted to investigate the source of intracellular $Ca^{2+}$ responsible for mitochondrial $Ca^{2+}$ overload. We tested whether $Ca^{2+}$ released from the ER, a major storehouse of intracellular $Ca^{2+}$, is the primary $Ca^{2+}$ source responsible for mitochondrial $Ca^{2+}$ overload. Most chemotherapeutic drugs cause ER stress, followed by $Ca^{2+}$ release that enters the mitochondria to stimulate apoptosis[45]. Roles of cisplatin in ER stress followed by release of ER-calcium are reported[46]. We first determined the effects of MICU1 silencing on the release of $[Ca^{2+}]_{ER}$ by cisplatin. CP20 cells with or without MICU1 silencing showed similar cisplatin-induced drop in $[Ca^{2+}]_{ER}$ (~ 50%) as determined by loss of fluorescence of Fluo-5N-AM (Fig. 4g). In addition, thapsigargin treatment that releases $Ca^{2+}$ from ER shows similar increase (~ 1.8-fold) in cytosolic $Ca^{2+}$ in both MICU1 silenced and control cells (Supplementary Fig. 12). Taken together, the results from Fig. 4g and Supplementary Fig. 12 suggest that MICU1 does not affect cisplatin-induced $[Ca^{2+}]_{ER}$ release and that the $Ca^{2+}$ storing capacity of ER- in MICU1-silenced and non-silenced cells are similar. Next we wanted to investigate whether ER-released $Ca^{2+}$ is responsible for cisplatin-induced apoptotic induction in MICU1 silenced cells. We pre-incubated cells with Pyr6 [(N-[4-[3,5-Bis(trifluoromethyl)-1H-pyrazol-1-yl]phenyl]-3-fluoro-4-pyridinecarboxamide) a

selective inhibitor for $Ca^{2+}_{ER}$ release] (ref. 47) before cisplatin treatment. Inhibition of $[Ca^{2+}]_{ER}$ release by Pyr6 abrogated cisplatin induced $[Ca^{2+}]_m$ overload in MICU1 silenced OvCa cells (Fig. 4h). Additionally, the cytotoxic effects of cisplatin (10 μM) were minimized in Pyr6 pretreated siMICU1-CP20 cells, suggesting a requirement for $[Ca^{2+}]_{ER}$ release by cisplatin, to execute anti-proliferative effects in MICU1 silenced cells (Supplementary Fig. 13). To establish the role of $[Ca^{2+}]_{ER}$ in cisplatin-induced toxicity we simultaneously treated cells with calcium chelator BAPTA-AM (10 μM) and cisplatin. In the presence of BAPTA-AM, MICU1 silenced cells did not show sensitization towards cisplatin-induced cell death, confirming the importance of influx of $[Ca^{2+}]_{ER}$ to the mitochondria for apoptotic induction (Supplementary Fig. 14). Taken together these results confirm that MICU1 restricts agonist (chemotherapeutics) stimulated entry of $Ca^{2+}$ into the mitochondria, and possibly impart chemoresistance in OvCa cells.

**Role of MICU1 in regulating OX-PHOS function.** Mitochondrial OX-PHOS and NADPH oxidases are the major ROS sources in most cells, while constitutively elevated $[Ca^{2+}]_m$ enhances basal bioenergetics[48]. Since we observed significant mtROS generation in MICU1 silenced cells, we next wanted to investigate whether MICU1 affects mitochondrial metabolism. The oxygen consumption rate (OCR) of cells was measured as an index of the mitochondrial electron transport[39] and the OCR is inversely related to aerobic glycolysis. Compared to the control

cells, ectopic expression of MICU1 in non-malignant cells (OSE and FTE188) significantly decreased the trifluoromethoxy carbonylcyanide phenylhydrazone (FCCP) induced stimulation of oxygen consumption (31 picomoles min$^{-1}$ for OSE and 22 picomoles for FTE188) (Fig. 5a–c) indicating potential role of MICU1 in reprogramming the glycolytic pathway towards aerobic glycolysis. To determine a role of MICU1 in mitochondrial respiration of OvCa cells, endogenous MICU1 was depleted (by siRNA) in CP20 and OV90 cells, and OCR was measured. Compared to the control cells MICU1 depleted cells significantly enhanced FCCP induced stimulation of OCR (73 picomoles min$^{-1}$ for CP20 and 98 picomoles min$^{-1}$ for OV90) (Fig. 5d–f), suggesting a role of MICU1 in maintaining aerobic glycolysis in OvCa cells. The observed FCCP stimulated response prompted us to explore whether MICU1 is primarily affecting the respiratory complexes in the electron transport chain

(ETC). We investigated effects of MICU1 silencing on the expression and the enzymatic activity of the ETC components. Any significant change in the expression of the key components of the mitocomplexes was not observed upon MICU1 depletion (Supplementary Fig. 15). Interestingly of the four ETC complexes, the complex III activity was increased in MICU1 silenced cells (Supplementary Fig. 16C) while the rest remain unchanged (Supplementary Fig. 16A,B,D). Indeed a high complex III activity is associated with generation of mtROS (ref. 49) and corroborates with our results demonstrating that MICU1 silenced cells show higher mtROS (Fig. 3e). The fact that MICU1 silencing increases mitochondrial respiration links MICUI1 to cellular energy demand and encouraged us to investigate whether MICU1 influences aerobic glycolysis (Warburg effect). Since glycolysis is implicated in chemoresistance in a variety of malignancies[50], we wanted to investigate whether cisplatin treatment of OvCa cells

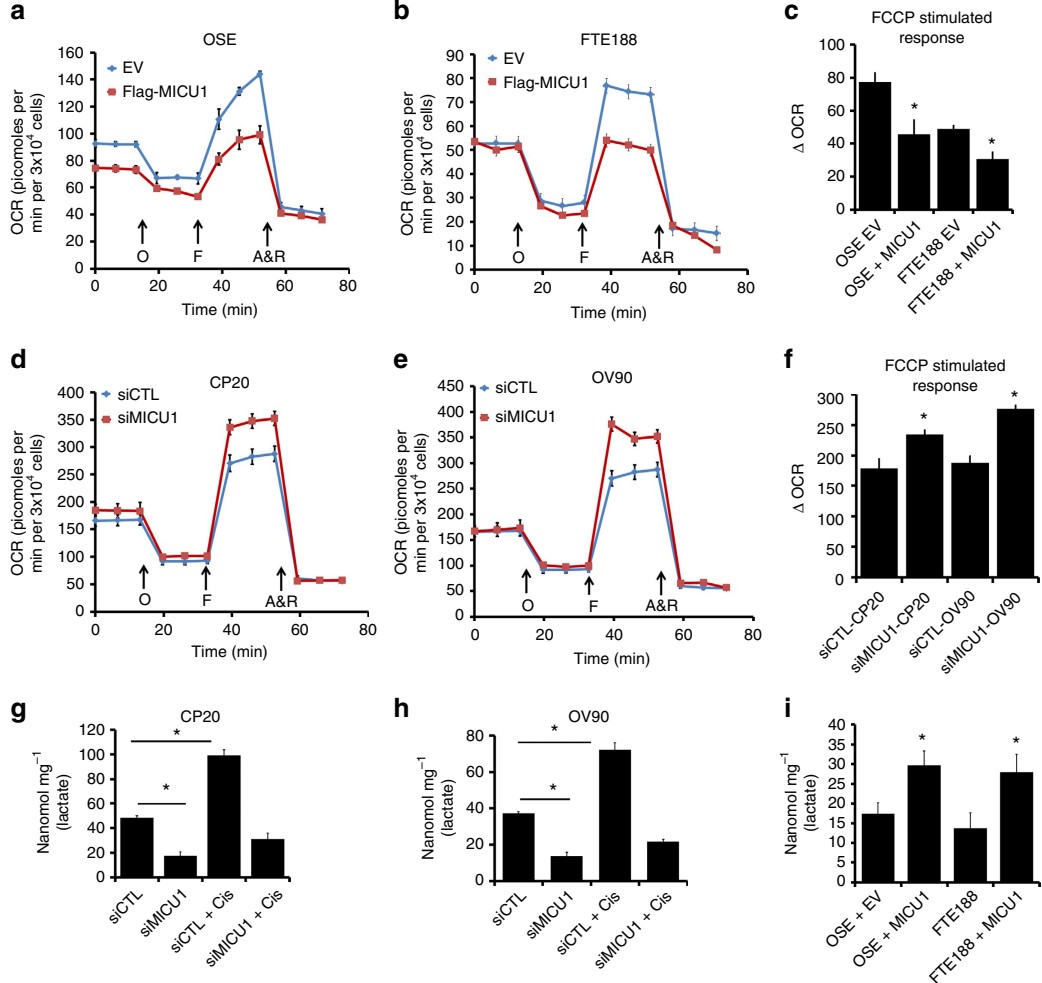

**Figure 5 | MICU1 negatively regulates OxPhos function and induces lactate production.** (**a,b**) Oxygen consumption rate (OCR) was measured using the Seahorse XF-96 analyser in OSE (**a**) and FTE188 cells expressing EV (empty vector) or FLAG-MICU1. An XF trace representing mean ± s.e.m. of at least three technical replicates per group is shown. Cells were sequentially treated with 2 μM Oligomycin (O), 0.5 μM FCCP (F) and 1 μM each of Antimycin + Rotenone (A + R) at indicated time points. (**c**) FCCP stimulated response were derived from the XF trace as in **a,b** from two independent experiments and represented as mean ± s.e.m. (**d**) CP20 and (**e**) OV90 cells were transfected with scrambled siRNA (siCTL) or MICU1 siRNA and the oxygen consumption rate (OCR) measured using the Seahorse XF analyser. An XF trace representing mean ± s.e.m. from at least three technical replicates per group is shown. (**f**) FCCP stimulated response were derived from the XF trace of two independent experiments and represented as mean ± s.e.m. (**g,h**) Intracellular lactate was measured in MICU1 silenced OV90 (**g**) and CP20 (**h**) cells with treatment with cisplatin 10 μM or vehicle and values represent mean ± s.d. (**i**) Intracellular lactate measurement in MICU1 overexpressing OSE and FTE188 cells as in **g**. *$P < 0.05$ statistically significant when compared to EV (**c,i**) siCTL (**f–h**). Comparisons between two groups were evaluated using two-sided Student's *t*-test with unequal variances. Horizontal bar represents statistical comparison between respective groups. All the experiments were repeated independently at least three times and in triplicates.

would induce glycolysis. We measured lactate production by both CP20 and OV90 cells with/without MICU1 silencing as a read-out for aerobic glycolysis (Fig. 5g,h). Without treatment with cisplatin intracellular lactate content in siCTL-CP20 cells was $\sim 48.367$ nmol mg$^{-1}$ while MICU1 silencing decreased lactate production in siMICU1-CP20 cells to $\sim 17.489$ nmol mg$^{-1}$. Similarly in OV90 cells the endogenous lactate concentrations dropped from $\sim 37.34$ nmol mg$^{-1}$ in siCTL-OV90 to 13.34 nmol mg$^{-1}$ in siMICU1-OV90 cells, indicating a reversal of Warburg effect. Furthermore, cisplatin treatment stimulated lactate production in both CP20 ($\sim 99.3$ nmol mg$^{-1}$ in siCTL-CP20) and OV90 $\sim 72.35$ nmol mg$^{-1}$ in siCTL-OV90) indicating OvCa cells utilize glycolysis to overcome the cytotoxic effects of cisplatin. Importantly, cisplatin-stimulated lactate production is inhibited upon MICU1 silencing (Fig. 5g,h) and lactate concentrations were nearly reduced to endogenous lactate levels as measured in siCTL-CP20 and siCTL-OV90. These results indicate that MICU1 functions as a glycolytic switch by preventing cisplatin induced $Ca^{2+}$ entry into the mitochondria and thereby preventing $[Ca^{2+}]_m$ overload. To further confirm a role of MICU1 as glycolytic switch, we determined the lactate content of normal OSE and FTE188 cells and compared with their MICU1 over expressing counterparts. MICU1 overexpression increased lactate concentrations in these normal cells (Fig. 5i) by almost twofold, further confirming a role of MICU1 as a glycolytic switch. Taken together, these results demonstrate that MICU1 functions as a metabolic switch that drives aerobic glycolysis and chemoresistance in OvCa.

**MICU1 alters PDH activation.** Cancer cells prefer a non-mitochondrial route for metabolism mainly through the aerobic glycolysis pathway (Warburg effect) to achieve fast and steady supply of energy and building blocks for rapid proliferation and survival[5]. To acquire this phenomenon, cancer cells reprogram the metabolic pathways[3]. In this context, $[Ca^{2+}]_m$ is known for its role in fine-tuning mitochondrial metabolic pathways by affecting activity of kinases and dehydrogenases[51]. The key checkpoint between aerobic glycolysis and mitochondrial TCA cycle is the PDH complex, which catalyses the conversion of pyruvate to acetyl-coA, a precursor of TCA cycle[52]. PDK phosphorylates PDH at ser293 and inhibits its enzymatic activity and thus shifts energy production towards aerobic glycolysis[53]. Since MICU1 is known to be a negative regulator of MCU and functions as a gate keeper of mitochondrial $Ca^{2+}$ uptake[17], we hypothesize that MICU1 silencing might stimulate pyruvate dehydrogenase phosphatases (PDPs) activity by increasing $Ca^{2+}$-overload in the mitochondria[53]. PDPs then can dephosphorylate PDH and render them active for mitochondrial oxidative phosphorylation and thereby switching energy requirement from glycolysis to TCA cycle. First, we determined the expression of pPDH(ser293), PDH, PDK and PDP in a panel of OvCa cell lines. We observed that majority of OvCa cells possess higher expression for pPDH as compared to non-malignant OSE cells (Fig. 6a), suggesting higher PDK activity in cancer cells irrespective of their expression levels. These results further suggest that glycolytic phenomenon to be a characteristic for most OvCa cells. To investigate a role of MICU1 in PDK-PDH-PDP axis we first determined any role of MICU1 in physical interaction between PDP and PDH. Indeed MICU1 silencing enhances association of PDP and PDH compared to siCTL cells by $\sim 58\%$, suggesting PDP driven activation of PDH (Fig. 6b,c). Further the pPDH expression was found to be downregulated in siMICU1 cells by $\sim 52\%$, while total PDH levels remained unaltered (Fig. 6d,e). Determination of PDH activity in MICU1-silenced CP20 (2.21 nmol min$^{-1}$ ml$^{-1}$) and

OV90 cells (1.98 nmol min$^{-1}$ ml$^{-1}$) demonstrated a significant increase in PDH activity when compared with the non-silenced scrambled controls (siCTL-CP20: 0.97 nmol min$^{-1}$ ml$^{-1}$ and siCTL-OV90: 1.32 nmol min$^{-1}$ ml$^{-1}$) (Fig. 6f). This increase in PDH activity in MICU1 silenced cells suggests a shift of aerobic glycolysis pathway towards mitochondrial glucose metabolism (TCA cycle). To further support the direct involvement of MICU1 in PDH inactivation by PDK, we ectopically overexpressed Flag-MICU1 in OSE cells. The PDP-PDH interaction was disrupted upon MICU1 overexpression in OSE cells as PDH failed to immunoprecipitate with PDP (Fig. 6g,h), a direct consequence of MICU1 driven lowering of $[Ca^{2+}]_m$. Furthermore, OSE cells harbouring Flag-MICU1 shows higher pPDH(ser293) expression (Supplementary Fig. 17) and decrease in PDH activity (0.81 nmol min$^{-1}$ ml$^{-1}$) as compared to empty vector transfected OSE cells (1.27 nmol min$^{-1}$ ml$^{-1}$) (Fig. 6i). In addition, treatment with cisplatin caused greater interaction of PDH and PDP in siMICU1-OV90 cells ($\sim 2$-fold) (Fig. 6j,k) in agreement with high $[Ca^{2+}]_m$ in cisplatin treated siMICU1-OV90 cells. Taken together, the above results indicate that MICU1 supports glycolytic phenotype by maintaining an inactive state of PDH. Since changes in $[Ca^{2+}]_m$ are also known to activate mitochondrial dehydrogenases[54], we investigated the expression levels of $\alpha$-ketoglutarate- and isocitrate-dehydrogenases (Supplementary Fig. 18A) and their enzymatic activities (Supplementary Fig. 18B,C) in MICU1 silenced CP20 or OV90 cells. We did not observe any significant changes neither in their expression levels nor enzymatic activities.

To further validate the mechanisms of MICU1 induced glycolysis in tumour growth and therapy resistance in patients, we determined expression of pPDH(ser293) and MICU1 in chemoresistant patient TMA using immunohistochemistry. Interestingly the expression of pPDH(ser293) is higher in chemoresistant HGSOC patient samples (Fig. 6l) and statistically significant correlation between MICU1 and pPDH (ser293) is observed in the chemoresistant TMA (Fig. 6l and Supplementary Table 2). The Spearman correlation coefficient between pPDH and MICU1 was significant among the various progression-free survival (PFS) groups ranging from 0.46 to 0.53 ($P < 0.05$, the SAS software version 9.3 was used to compute the $P$ values.). This clinical analysis supports the concept that MICU1, by restricting $Ca^{2+}$ entry into the mitochondria and thereby attenuating PDP activity, drives PDK-dependent phosphorylation of PDH in these tumours (Fig. 6h) and thereby promotes aerobic glycolysis. Furthermore, high pPDH expression is significantly correlated with low PFS, indicating aerobic glycolysis is associated with poor prognosis in OvCa patients (Supplementary Table 3). All of the experimental evidences support that MICU1 is responsible for deranged metabolism in OvCa by rewiring the PDK-PDH-PDP nexus and plays an important role in poor prognosis in OvCa.

**Role of MICU1 in tumour growth and drug sensitivity *in vivo*.** The unavailability of any specific pharmacological inhibitor for MICU1 posits a challenge to establish roles of MICU1 in chemosensitization and tumour growth *in vivo*. To overcome the aforesaid challenge and confirm a role of MICU1 in tumour growth and therapy resistance we performed xenograft studies using the shCTL- and shMICU-OV90 cells. We implanted shCTL or shMICU1 cells intrabursally into the ovaries of 6–8-week-old athymic nude female mice and monitored tumour growth over a 4-week period. Since *in vitro* silencing of MICU1 pruned OvCa cells towards chemosensitization by cisplatin, therefore, we wanted to investigate whether a low dose of cisplatin could effectively inhibit ovarian tumour growth. For 3 weeks the mice received i.p. cisplatin (160 µg per mouse) or Hanks' Balanced Salt

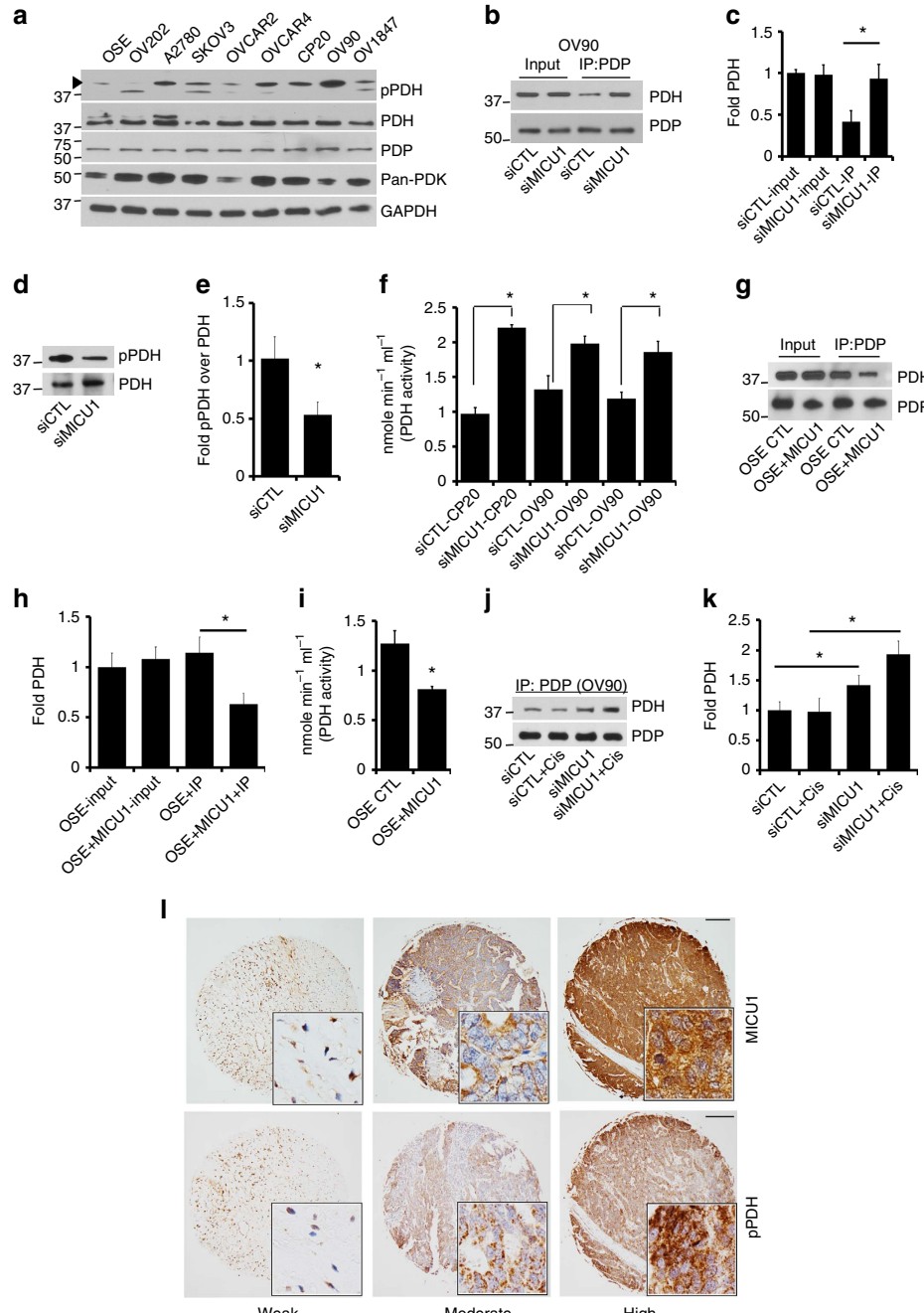

**Figure 6 | MICU1 silencing restores pyruvate dehydrogenase activity.** (**a**) Immunoblot analysis for phospho pyruvate dehydrogenase (pPDH[ser293]), pyruvate dehydrogenase (PDH), pyruvate dehydrogenase phosphatase (PDP) and pan-pyruvate dehydrogenase kinase (pan-PDK) in OSE and various OvCa cell lines. GAPDH was used as a loading control. (**b**) MICU1 silencing increases PDP–PDH interaction. Immunoprecipitation of PDP followed by immunoblot for PDH and PDP in siCTL-OV90 and siMICU1-OV90 cells. (**c**) Quantification for PDH band intensity measurement using ImageJ software normalized against PDP. Values represent mean ± s.d. (**d**) MICU1 silencing causes diminished phosphorylation status of PDH. Immunoblot for pPDH in whole cell lysates from siCTL-OV90 and siMICU1-OV90 cells and compared against PDH. (**e**) Densitometry analysis of (**d**) by ImageJ, pPDH band intensity is normalized to corresponding PDH levels and represented as mean fold change ± s.d. (**f**) Pyruvate dehydrogenase (PDH) activity was measured in MICU1 siRNA treated OV90 or CP20 cells and MICU1 knockdown OV90 cells and values represent mean ± s.d. (**g**) MICU1 overexpression decreases PDP and PDH interaction. Immunoprecipitation of PDP followed by immunoblot for PDH and PDP in Flag-MICU1 expressing OSE cells. (**h**) Quantification for PDH band intensity measurement using ImageJ software normalized against PDP and values represent mean ± s.d. (**i**) In Flag-MICU1 expressing OSE cells, PDH activity was measured and values represent mean ± s.d. (**j**) MICU1 silencing increases PDP and PDH interaction upon cisplatin (10 μM) treatment. Immunoprecipitation of PDP followed by immunoblot for PDH and PDP in siCTL-OV90 and siMICU1-OV90 cells. (**k**) Quantification for PDH band intensity measurement using ImageJ software normalized against PDP. (**l**) Immunohistochemical staining of a tissue microarray of drug resistant epithelial OvCa samples for MICU1 (1:150) or pPDH (1:200). Representative images taken at × 4 magnification are shown of (i) weak, (ii) moderate and (iii) high staining. Inset shows magnified areas of individual IHC stains taken at × 20 magnification. The bar represents 100 μm. *$P < 0.05$ statistically significant when compared to the respective controls, using two-sided Student's $t$-test with unequal variances. Horizontal bar represents statistical comparison between respective groups. All the experiments were repeated independently at least three times and in triplicates.

Solution (HBSS) twice weekly. On day 30, the mice were killed and the tumours were collected for further analysis. We observed that the shCTL-implantation group formed statistically significant ($P < 0.05$) larger tumours (Mean tumour weight 1.469 g) than the shMICU1 group (Mean tumour weight: 0.353 g). Compared to HBSS treated control group, cisplatin treatment resulted in a significant reduction of tumour growth in shMICU1 group (mean tumour weight 0.169 g). It is also important to note that although the tumour growth inhibition by cisplatin seems to be proportionally identical between shCTL and shMICU1, in the shMICU1 group four out of five animals showed negligible tumour formation upon cisplatin treatment (80% of the sample size) as compared to one out of five animals in shCTL + cisplatin group (20% of the sample size), supporting a role of MICU1 in drug resistance. However, we do not rule out the possibility of other pro-oncogenic function of MICU1 that is currently unknown but will evolve over time. The health of the mice in each group was monitored daily and the dose schedule for each group appeared to be non-toxic. We further investigated the effect on overall survival of the animals bearing shCTL- and shMICU1-OV90 cells with/without cisplatin. Since the formation of large tumours in shCTL group animals turned them moribund and appeared in distress and hence were killed all on the same day according to the institutional animal care and use committee (IACUC) approved protocol. While animals bearing tumours from shCTL and shCTL + cisplatin cells had median survival of 30 and 32 days respectively, shMICU1 and shMICU1 + cisplatin cells showed significant longer median survival of 43 and 53 days respectively ($P < 0.05$) (Fig. 7b). Thus, MICU1 silencing increases survival in animals bearing OvCa tumours suggesting targeting MICU1 in OvCa could be a strategy to prolong survival in patients. Growth regression was further confirmed by quantifying the number of proliferating cells using Ki67 staining (Fig. 7c). A significant decrease in Ki67 staining was observed in shMICU1 ($\sim$64% reduction) and shMICU1 + cisplatin ($\sim$74% reduction) group as compared to the shCTL (Fig. 7d). These results were further supported by the observation that both cisplatin-treated shMICU1 and shMICU1 + HBBS tumours showed a significantly large increase in number of TUNEL positive nuclei ($\sim$1.5-fold and $\sim$2.43-fold respectively) than the cisplatin-treated shCTL or shCTL + HBSS tumours (Fig. 7e), indicating significant induction of apoptosis in shMICU1 groups (Fig. 7f). All of this data convincingly demonstrate that MICU1 is a key player for tumour growth and therapy resistance and targeting MICU1 sensitizes OvCa cells to cisplatin both *in vitro* and *in vivo*. Furthermore, protein expression for the inactive pPDH(ser293) and lactate producing enzyme LDH was lower in tumours from shMICU1 cells as compared to those from shCTL cells (Fig. 7g), suggesting prevalence of reduced aerobic glycolysis in tumours from shMICU1 animals. To further support a role of MICU1 in promoting glycolysis *in vivo*, we determined the effect of MICU1 silencing on the expression and enzymatic activity of PDH in tumour tissue samples. The PDH activity was twofold higher in tumour tissues from shMICU1 animals (Fig. 7h) compared to those from shCTL animals, suggesting lowering of aerobic glycolysis. Thus, our preclinical results further confirm that MICU1 drives aerobic glycolysis and chemoresistance in OvCa.

## Discussion

Epithelial OvCa is one of the deadliest gynaecological malignancies of women in the western world. Epithelial OvCa is characterized by frequent relapse and evolution of drug resistance in majority of the patients with advanced stage disease[55,56]. Previously Fabian *et al.* demonstrated that the tumorigenic property of OvCa cells was dependent on their glycolytic phenotype, cells such as OC316 demonstrating higher glycolytic phenotype was more aggressive compared to IGROV-1 cells with cells glycolytic property[57]. Therefore, interrogating metabolic aberrations particularly glycolysis in OvCa cells may potentially be an effective therapeutic strategy to improve poor outcome.

It has been recognized that glycolysis plays an important role in tumour growth and therapy resistance[7]. Several glycolytic pathway enzymes are currently being explored as potential chemotherapeutic targets[1,58–62].

In this context, we identified a novel role for MICU1 in maintaining a glycolytic phenotype in OvCa cells. Since MICU1 is a negative regulator of mitochondrial $Ca^{2+}$ uptake silencing of which causes mitochondrial $Ca^{2+}$ overload[63], we hypothesize that MICU1 plays a critical role in regulating functions of some of the $Ca^{2+}$ dependent metabolic enzymes responsible for deranged metabolism. The hypothesis is further supported by the fact that three matrix dehydrogenases are activated by $Ca^{2+}$, such as: PDH which is regulated by a $Ca^{2+}$-dependent phosphatase, while α-ketoglutarate- and isocitrate-dehydrogenases are regulated by direct binding of $Ca^{2+}$ to these enzymes[64,65]. The enzyme PDH in its phosphorylated (ser293) form remains inactive, and promotes aerobic glycolysis as it fails to funnel the pyruvate to mitochondrial oxidation pathway[66]. Indeed, we observed higher pPDH expression in our chemoresistant patient TMA which strongly correlates with lower PFS and MICU1 expression. The PDH is phosphorylated by the kinase PDK at ser293 and the later serves as a prognostic marker in many cancer types[67]. We determined a clinical correlation between MICU1 expression and overall survival in two separate clinical cohorts. The analysis demonstrated that high *MICU1* expression was correlated with poor OS in these cohorts. These clinical correlations further define MICU1 as a metabolic switch that we delineated using cell culture model.

Recently Mallilankaraman *et al.*[35] have hypothesized MICU1 activity to be involved in adaptation to toxicity and greater survival under ceramide stress in HeLa and HMEC cells[35]. Our findings in OvCa cells and preclinical models demonstrate that chemosensitization can be achieved by targeting MICU1. Silencing of MICU1 resulted in $Ca^{2+}$ overload in the mitochondria, a prerequisite for apoptosis and concomitant cisplatin exposure resulted in efficient apoptotic cell death. A recent publication by Liu *et al.* demonstrated in mouse model of MICU1 deficiency [MICU1$^{(-/-)}$] that MICU1 acts as a molecular gatekeeper to prevent *in vivo* mitochondrial calcium overload and absence of MICU1 is associated with increased resting mitochondrial calcium levels, altered mitochondrial morphology and reduced ATP (ref. 68). In this context, MICU1 has also been shown to be involved in survival and tissue regeneration in liver and loss of MICU1 is associated with impaired cell cycle entry, $Ca^{2+}$ overload-induced mitochondrial permeability transition pore opening is accelerated in MICU1-deficient hepatocytes and extensive necrosis. Summarily, our work and recent findings identify an unanticipated role of MICU1 in survival and highlights the importance of regulating MICU1 under stress conditions when the risk of $[Ca^{2+}]_m$ overload is elevated[69].

Present study defines a role of MICU1 in regulating drug resistance phenotype in cancer model. Our finding is further supported by two recent papers from Zhou *et al.*[70] and Tian *et al.*[71] that identified EZH2 and RSP3 as prognostic markers for HNSSC and melanoma, respectively. Targeting both EZH2 and RSP3 resulted in apoptotic cell death in their respective models with a common denominator of diminished MICU1 expression during apoptosis[70,71]. Another phenotype determined for MICU1 was its ability to affect clonal growth of OvCa cells, which was abrogated upon MICU1 silencing. Clonal growth represents

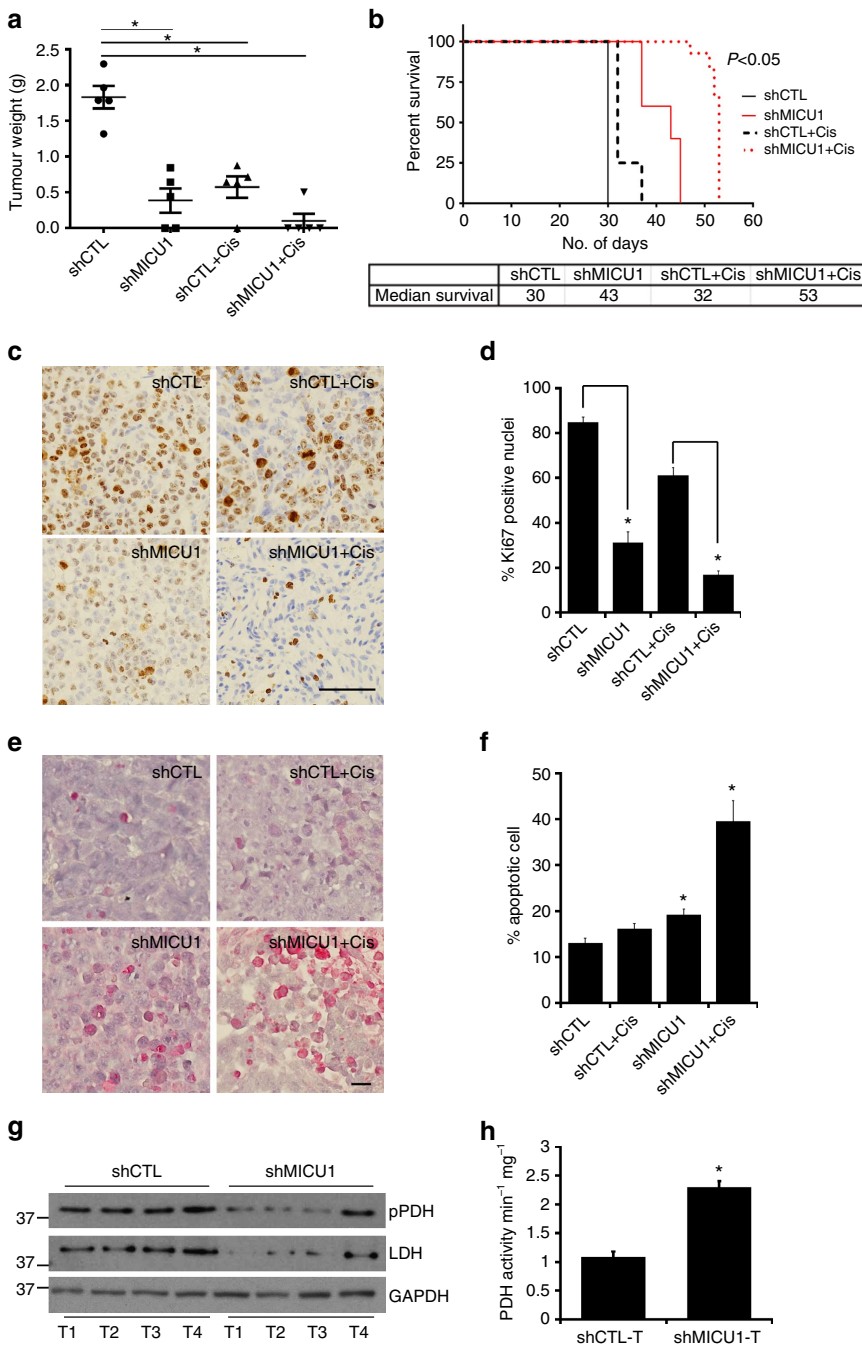

**Figure 7 | Effect of MICU1 knockdown on tumour growth and drug resistance.** (**a**)To assess the effects of knockdown of MICU1 on tumour growth, treatment with cisplatin was initiated 1 week after i.p. injection ($1.0 \times 10^6$ shCTL- or shMICU1-OV90) of tumour cells. Mice were divided into four groups ($n = 10$ mice per group): (i) shCTL, (ii) shCTL + cisplatin (160 μg per mouse i.p. twice weekly), (iii) shMICU1 and (iv) shMICU1 + cisplatin (160 μg per mouse i.p. twice weekly). Treatment was continued until 4 weeks after tumour inoculation before the killing. (**a**) Scattered plot of tumour weight ($n = 5$) in HBSS-treated (sham) or 160 μg per mouse i.p. cisplatin-treated animals post 30 days of treatment. Statistical analysis was performed using one-way ANOVA followed by Newman-Keuls multiple comparison test. (**b**) Kaplan-Meier curves were plotted for the four groups of animals ($n = 5$). Log-rank test $P$ value was reported. (**c**) Representative histology of tumours from mice xenografts of shCTL-OV90 or shMICU1-OV90 cells with Ki67 expression and acquired at ×20 magnification. Scale bar represents 100 μm. (**d**) Quantification of Ki67 staining in the mouse xenografts respectively and values represent mean ± s.d. (**e**) Representative histology of tumours from mice xenografts of shCTL-OV90 or shMICU1-OV90 cells with TUNEL staining and acquired at ×20 magnification. (**f**) Quantification of TUNEL staining in the respective mouse xenografts and values represent mean ± s.e.m. (**g**) Immunoblotting for the expression of glycolytic enzyme lactate dehydrogenase (LDH) and pPDH(ser293) in tissues from mice tumour from shCTL-OV90 and shMICU1-OV90 cells. GAPDH was used as a loading control. (**h**) PDH activity measured in tumour tissues from shCTL-OV90 and shMICU1-OV90 cells and values represent mean ± s.d. *$P < 0.05$ was statistically significant when compared to the respective controls, using two-sided Student's $t$-test with unequal variances (**d,h**) or ANOVA with Bonferroni's correction (**f**). Horizontal line represents statistical comparison between respective groups. All the experiments were repeated independently at least three times and in triplicate.

'stemness' of the cell[30] and depends on multiple signalling cascades like non-canonical Wnt or BMP pathway and each of them can be modulated by deregulated $Ca^{2+}$ homeostasis[72].

It is now well accepted that most cancers have a glycolytic phenotype[3]. Our data suggest that this apparent mitochondrial 'dysfunction' is in fact reversible and can be normalized to mitochondrial oxidative metabolism and implicate a novel way to reverse chemoresistance by regulating expression of a mitochondrial $Ca^{2+}$-uptake protein. In summary, our work identifies MICU1 as a critical component responsible for aberrant metabolism that fuels aerobic glycolysis and chemoresistance. Thus, MICU1 may be exploited as an important therapeutic target to normalize deranged metabolism to inhibit tumour growth and overcome therapy resistance.

## Methods

**Reagents and cell culture.** The information for the source and working dilution of the antibodies used are provided in Supplementary Information Table 4. Rhod-2AM (#ENZ-52010) was from Enzo Life Sciences, Pyr6 (#SML1241) was from Sigma (St. Louis, USA) and MitoSOX Red mitochondrial superoxide indicator (#M36008) Fluo-5N,AM (#F-14204) were procured from Invitrogen (Carlsbad, USA). MitoTEMPO (#SML0737), Lactate assay kit (#MAK064) and PDH assay kit (#MAK183) were purchased from Sigma (St. Louis, USA) and TMRE-Mitochondrial Membrane Potential Assay kit (#ab113852) was procured from Abcam (Cambridge, USA). Three Step Staining Set for was from Thermo Scientific (#3300). Cell culture media RPMI-1640 and 0.25% Trypsin-EDTA were from Lonza. MCDB105 and Medium199 were from Sigma-Aldrich. Lipofectamine 3000 and OPTIMEM were from Invitrogen. CP20, A2780, OSE and OV90 cells were kindly gifted by Dr Anil Sood, MD Anderson Cancer Center. OVCAR4 was a kind gift from Dr Ronny Drapkin, formerly at the Dana-Farber Cancer Institute, Boston, MA, USA. Immortalized normal fallopian tube epithelial cells (FTE-188) were a kind gift from Dr Jinsong Liu, MD Anderson Cancer Center. OVCAR2 and OV1487 were a kind gift from Dr Susan K. Murphy, Duke University, Durham, NC, USA. TYKNU, TYKNUcis and Kuramochi were purchased from JCRB Cell Bank, Japan. SKOV 3 was purchased from ATCC. All the cell lines were routinely cultured in RPMI (OV90, CP20, OVCAR4, OVCAR2, OV1487 and Kuramochi); ATCC-EMEM (TYKNU and TYKNUcis) and 1:1 Media 199: MCDB 105 (OSE and FTE188). Media was supplemented with 10–15% FBS and routinely tested for mycoplasma contamination by PCR-based method.

**Plasmids and siRNA transfection.** Gene silencing was performed for CP20 and OV90 cell lines in 6 cm culture dish containing $5 \times 10^5$ cells using Hiperfect (Qiagen) and commercially validated 10 μM siRNA (scrambled control siRNA (#1027280), QIAGEN, CA, USA) or siRNA against human MICU1 (SASI_Hs01_00070243, SASI_Hs01_00070249 Sigma) (# S1041403600 Qiagen) in OPTIMEM (Invitrogen). Effective silencing was achieved after 48–72 h of transfection (determined by protein expression) and all experiments with gene silencing were performed 48–96 h post transfection. Transfection of OSE and FTE188 cells with Flag-MICU1 (pLYS1-MICU1-Flag was a gift from Vamsi Mootha, Addgene plasmid # 50058) (ref. 73) or Empty vector was performed using Lipofectamine 3000 with 4 μg plasmid DNA. Cells were incubated for 24 h after transfection before testing for transgene expression or performing downstream experiments.

**Generation of stable MICU1 knockdown cell lines.** For generating stable cell lines, control shRNA and shRNA targeting MICU1 (shMICU1) in lentiviruses were from Sigma. Resultant cell lines were selected with 2 μg ml$^{-1}$ puromycin for 2 weeks. OV90 cells ($5 \times 10^5$ per well) grown in six-well plates were transduced with lentiviruses harbouring shMICU1, selected with puromycin (2 mg ml$^{-1}$) 48 h post transduction for 6–10 days and expanded. Knockdown was assessed by qRT-PCR and western blotting.

**Immunoblotting.** Immunoblotting analysis was carried on cell lysates in RIPA buffer supplemented with protease-phosphatase mix (Pierce). Briefly, the cell lysate was separated on 10% or 12% tris-glycine SDS-PAGE gel and transferred to PVDF membrane. Membranes were blocked in 5% non-fat dry milk in TBS with 0.1% TWEEN-20 (TBST) for 1 h at room temperature followed by overnight incubation with indicated primary antibodies in TBST with 5% BSA. Primary antibodies were used in 1:1,000 dilutions and secondary antibodies were used at a concentration of 1:10,000 for 1 h a room temperature. Equal loading was verified by immunoblotting with GAPDH, VDAC or actin. Immunoblotting images were scanned and quantified with ImageJ (image processing and analysis in Java, NIH) using the loading control to normalize the experimental value. Uncropped images of the important immunoblots are provided in Supplementary Fig. 19.

**Immunoprecipitation.** Immunoprecipitation was performed using Pierce Crosslink Immunoprecipitation Kit as per manufacturer's protocol. Briefly, 10 μg antibody was covalently crosslinked with agarose A/G beads using DSS and incubated with 1 mg of total cell lysate, overnight at 4 °C. Antigen was eluted and subjected to SDS page electrophoresis.

**Soft agar assay.** Forty-eight hours post siMICU1 transfection $1 \times 10^3$ CP20 or OV90 cells in RPMI medium containing 10% FBS with 0.3% agar (SeaPlaque GTG Agarose, Lonza Allendale, NJ 07401, USA) were seeded on top of 0.6% agar in the same medium in each well of 12-well plate. After 10-14 days, the colonies were stained with 0.1% crystal violet, imaged using Leica EZ4HD (Buffalo Grove, IL 60089, USA). Colonies were imaged in a blinded fashion and nine images from three independent experiments were quantified using ImageJ.

**Cell mito-stress analysis.** Different parameters of mitochondrial respiration was measured with the 96-well XF analyser and the XF cell mito stress kit (Seahorse Bioscience, MI, USA). Assay was done as per the manufacturer's protocol. Twelve hours before assay, XF sensor cartridges were hydrated with the supplied calibrant and incubated at 37 °C without $CO_2$ (pH to 7.4 at 37 °C). Unbuffered XF assay medium was prepared and transfected cells post 48 h for siRNA and 24 h for Flag-MICU1 were re-plated from a 60-well dish to the 96-well XF assay plate at a cell density of $3 \times 10^4$ cells per well. Thirty minutes before the assay, the plates were washed with the assay medium and incubated at 37 °C without $CO_2$. Oligomycin (2 μM), FCCP (500 nM) and a mix of Antimycin A and Rotenone (1 μM each) were added into the appropriate ports of the cartridge and calibrated in the instrument. After the calibration of the cartridge, cell culture plate was loaded in the instrument and the assay was run as per the standard template and OCR was measured.

**PDH Activity and lactate measurement.** PDH activity was measured using a kit (#MAK183) from Sigma as per the manufacturer's instruction. Briefly, $5 \times 10^5$ cells were seeded in a 35 mm plate and allowed to grow for 48 h. Cells well lysed in assay buffer and 5-10 μl of cell lysate were added to wells of 96-well plates. Appropriate reaction mix consisting of assay buffer, substrate and developer was added to each of the wells and the product of coupled enzyme reaction, which results in a colorimetric (450 nm) product proportional to the enzymatic activity. The absorbance was recorded continuously by incubating the plate at 37 °C taking measurements (450 nm) every 5 min for 40 min. Lactate concentration was determined by using a kit (#MAK064) from Sigma as per the manufacturer's instruction. Briefly, $5 \times 10^5$ cells were seeded in a 35 mm plate and allowed to grow for 48 h and then lysed in lactate assay buffer. Aliquots of lysate 5-10 μl were distributed in 96-well plates suitable for absorbance measurement. Appropriate reaction mix consisting of assay buffer, substrate and developer was added to each of the wells. In this assay, lactate concentration was determined by an enzymatic assay, which results in a colorimetric (570 nm) product, proportional to the lactate concentrations.

**Mitochondrial membrane potential assay.** Mitochondrial membrane potential was measured using a kit (#ab113852) from Abcam as per the manufacturer's instruction. Briefly, $5 \times 10^5$ cells were transfected as described and 48 h post transfection 10,000 cells were replated in a 96-well plate suitable for fluorescence measurement. positively charged TMRE (tetramethylrhodamine, ethyl ester) dye (250 nM) was used to label active mitochondria that are negatively charged. FCCP, Carbonyl cyanide p-trifluoro-methoxyphenyl hydrazone, an ionophore that depolarizes mitochondria and prevents binding of the dye to the mitochondria is used as a negative control for the assay. Plate was read at 549 nm excitation and 575 nm emission wavelengths.

**Proliferation assays.** Transfected OV90 or CP20 cells were collected by trypsinization after 48 h of transfection, counted and seeded in 96-well plates ($2.5 \times 10^3$ cells per well) and cultured for 24 h. Cell proliferation was determined using the CyQUANT NF Cell Proliferation Assay Kit (Invitrogen, C7026) according to the manufacturer's protocol and fluorescence intensity was measured at excitation at 485 nm and emission detection at 530 nm. Experiments were repeated at least three times each time in triplicate. Additionally, BrdU incorporation assay was performed using the Cell Proliferation ELISA, BrdU Kit (Roche). Transfected OV90 or CP20 cells were collected by trypsinization after 480 h of transfection, counted and seeded in 96-well plates ($3 \times 10^3$ cells per well) and cultured for 24 h. Post cisplatin treatment, cells were incubated with BrDU for 12 h, fixed, probed with antibody supplied in kit and colorimetric signals were measured at 450 nm.

**Boyden chamber migration and invasion.** Post 48 h transfection with scrambled siRNA or MICU1 siRNA, OV90 or CP20 cells were serum-starved overnight, detached from culture plates by trypsinization and $1 \times 10^5$ cells were plated into 8 μm transwell chambers in 200 μl of serum-free RPMI1640 medium. The lower chambers of the plate were supplied with 650 μl of RPMI1640 medium with 10%

FBS. The cells were allowed to migrate for 12 h after which cells were processed with Three Step Stain Set. Cells in the upper chamber were removed using a cotton swab, and cells migrating through the membrane were counted. Cell invasion studies were performed using Boyden chamber equipped with membranes pre-coated with 100 $\mu$g ml$^{-1}$ fibronectin (#F1141, Sigma).

**Gelatin degradation assay.** Acid-washed coverslips were first coated with 50 $\mu$g ml$^{-1}$ poly-L-lysine for 20 min at room temperature, and then fixed with 0.5% glutaraldehyde for 15 min. Gelatin matrix was prepared by mixing 0.2% gelatin and Oregon Green 488 Gelatin Conjugate (Life Technologies, Rockford, IL, USA) at an 8:1 ratio. After coating for 10 min, coverslips were washed with PBS and quenched with 5 mg ml$^{-1}$ sodium borohydride for 15 min followed by washing with PBS. For degradation assay, 20,000 cells were seeded in each well of a 24-well plate containing acid washed cover slips. Forty-eight hours after cell seeding, cells were fixed in 4% paraformaldehyde and stained with Alexa Fluor 555 Phalloidin (Life Technologies, Rockford, IL, USA) for 15 min at room temperature. The cells were washed with PBS and mounted with VECTASHIELD mounting medium containing DAPI (Vector Laboratories). Images were acquired at $\times 40$ using the Zeiss Axio-Observer Z1 (Göttingen, Germany). Cells that degraded the ECM were scored as positive from three independent experiments and were quantified.

**RNA isolation and analysis of gene expression by RTqPCR.** RNA extraction was performed using commercial kit (RNeasy Plus Mini kit, QIAGEN) following manufacturer's protocol. Twenty microliter reverse transcription reaction mix was prepared with Reverse Transcription Kit (iScript cDNA Synthesis kit, Bio-Rad) using equal volume of random primers and oligo dT, and 2 $\mu$g (for cell pellet) or 200 ng (for mitochondrial pellet) of isolated RNA. Quantitative real-time RT-PCR analysis was carried out in a 10 $\mu$l reaction volume including 1 $\mu$l cDNA template (diluted 20 times), SYBR Green Master Mix (Applied Biosystems) and IDT primers as listed below. The comparative $C_t$ method was used to calculate the relative abundance of the mRNA and compared with that of GAPDH or 36$\beta$4.

MCU-fw: 5′-GCAGAATTTGGGAGCTGTTT-3′
MCU-rv: 5′-GTCAATTCCCCGATCCTCTT-3′
MICU1-fw: 5′-GAGGCAGCTCAAGAAGCACT-3′
MICU1-rv:5′-CAAACACCACATCACACACG-3′
MICU2-fw: 5′-GGCAGTTTTACAGTCTCCGC-3′
MICU2-rv: 5′-AAGAGGAAGTCTCGTGGTGTC-3′
GAPDH-fw: 5′-CACCATCTTCCAGGAGCGAG-3′
GAPDH-rv: 5′-CCTTCTCCATGGTGGTGAAGAC-3′

**Calcium measurement.** Mitochondrial $Ca^{2+}$ measurements were carried out using Rhod-2 AM, which contains a rhodamine-like fluorophore. Cells were transfected as described and 48 h post transfection 10,000 cells were replated in a 96-well plate suitable for fluorescence measurement. Cells in HBSS (without $Ca^{2+}$ and $Mg^{2+}$) were loaded with 5 $\mu$M Rhod-2 AM and incubated for 10 min at 37 °C, washed with HBSS and the fluorescence was recorded at excitation 549 nm/emission 578 nm. For determination of $[Ca^{2+}]_{ER}$, Fluo-5N,AM (5 $\mu$M) dye was loaded to cells and the fluorescence was recorded at excitation 494 nm/emission 516 nm. To assess the mitochondrial $Ca^{2+}$ dynamics, cells were transiently transfected with fluorescent protein-based mitochondrial $Ca^{2+}$ indicator, GCaMP2-mt (ref. 74). At 48 h post-transfection, cells were challenged with cisplatin, and GCaMP2 fluorescence was measured using a Zeiss Axiovert200 equipped with a high definition imaging (HDI) scientific CMOS camera driven by the EasyRatioPro software (photon Technology International). Excitation was set at 475 nm and emission was set at 515 nm. Data are expressed as $F/F_0$, where $F$ is the fluorescence at any given time point and $F_0$ is the average fluorescence before the addition of cisplatin.

**Measurement of mitochondrial superoxide (mtROS).** Mitochondrial superoxide was measured using the mitochondrial superoxide indicator MitoSOX Red (Molecular probes; Invitrogen). Briefly, $3 \times 10^3$ cells grown on coated glass coverslips were loaded with 5 $\mu$M MitoSOX Red for 30 min, coverslips were mounted and images were acquired using Olympus FV-500 microscope at 561-nm excitation using a $\times 40$ water objective.

**Preclinical model of OvCa.** Female athymic nude mice (NCrnu; 5–6-week-old) were purchased from the National Cancer Institute—Frederick Cancer Research and Development Center. Xenograft studies were carried out on 6–8- week-old female mice. All mice were housed and maintained under specific pathogen-free conditions in facilities approved by the American Association for Accreditation of Laboratory Animal Care and in accordance with current regulations and standards of the US Department of Agriculture, US Department of Health and Human Services and National Institutes of Health. All studies were approved and supervised by the OUHSC Institutional Animal Care and Use Committee. A total of 40 animals were divided into two groups, each receiving $1 \times 10^6$ shCTL-OV90 or shMICU1-OV90 cells in 50 $\mu$l HBSS into the bursa of the ovary. The two groups ($N = 20$) were randomized and categorized based on the treatment regime ($N = 10$) as: (i) shCTL + HBSS, (ii) shCTL + 160 $\mu$g cisplatin, (iii)

shMICU1 + HBSS and (iv) shMICU1 + 160 mg cisplatin. Post 7 days of implantation, respective animals received biweekly intraperitoneal injections of 100 $\mu$l HBSS or 160 $\mu$g of cisplatin in 100 $\mu$l volume of HBSS. Treatment was continued for 21 days. After 4 weeks of implantation, the animals were killed and the tumours and tissues were collected for further analyses.

**Immunohistochemistry.** Tumour grafts or mouse tissues were embedded in paraffin and sectioned (4 $\mu$m). These sections were deparaffinized in xylene, rehydrated in graded alcohol, subjected to heat-induced antigen retrieval with Target Retrieval Solution, and blocked with Protein Block. Immunohistochemistry was performed according to standard protocols. Antigen retrieval was achieved by heating sections in 95 °C citrate buffer for 10 min. Sections were incubated with specific antibodies overnight at 4 °C. For Ki67 (1:100) staining, the dark brown signal was revealed after incubation with the ABC kit (Vector), followed by a diaminobenzidine and hydrogen peroxide reaction using the diaminobenzidine detection kit (Vector). Counterstaining was performed by incubating the slides in hematoxylin for 5 min. Images were taken using Nikon Eclipse Ni microscope. Tunnel staining was performed after deparafinization of 4 $\mu$m sections using In situ Cell Death Detection Kit, AP (Roche Diagnostics GmbH, Manheim, Germany) following manufacturer's protocol.

**mRNA analyses in two OvCa cohorts.** We used clinically annotated mRNA expression data from GEO databases. mRNA expression data for all genes were detected by microarray. The clinicopathologic and platform information of the two cohorts can be found in Supplementary Table 1. Log-rank test were used to evaluate the association of gene expression with patient survival. The upper-quantiles were used to divide the data into upregulate (red) and downregulated (green) groups for each gene. Kaplan-Meier curves were plotted for all genes, with a $P$ value < 0.05 considered significant.

**Tumour microarray analysis.** The TMA is from a patient population that includes 25 platinum refractory (not achieving a complete clinical response (CR) following primary therapy), 50 platinum resistant patients including 25 patients who achieved a CR, then recurred within 6 months from last use of platinum (platinum resistant) and 25 patients who achieved a CR, then recurred > 6 months but < 12 months from last use of platinum (platinum intermediate group), and 50 patients with platinum sensitive disease, defined as having achieved a CR, then recurred > 12 months from last use of platinum primary therapy. Written informed consent was obtained from all women enrolled into the study and Institutional Review Board approval was provided by OUHSC.

Immunohistochemical staining was done using MICU1 antibody (1:150) and pPDH antibody (1:200) and based on immunoreactivity and intensity, staining was graded as 1 (weak), 2 (moderate) and 3 (high). For statistical analysis for correlation of MICU1, pPDH and PFS, patients were categorized into three groups based on PFS: < 6 months, 6-12 months and > 12 months. Data were summarized using mean (s.e.m.). The correlation between pPDH and MICU1 within each PFS group was assessed by Spearman's correlation coefficient ($\rho$). The expression levels of pPDH and MICU1 were compared among the three PFS groups using the Kruskal-Wallis test. If this overall test was significant, pairwise comparisons were further carried out, adjusting for multiple comparisons based on Bonferroni's method. The association between the molecule (pPDH or MICU1) and OS within each PFS group was assessed using the Cox model. Hazard ratio (HR) and 95% confidence interval (CI) were reported. A two-sided $P$ value of < 0.05 defines statistical significance. SAS 9.3 software was used for data analysis.

**Data analysis and statistics.** All grouped data are presented as mean ± s.d. or mean ± s.e.m. and statistical significance was determined by using a two-tailed Student's $t$-tests. For the survival analysis, Kaplan–Meier curves were generated using Prism software and log-rank analysis were performed. All the experiments were repeated independently at least three times and in triplicate. A $P$ value of < 0.05 was defined statistically significant.

**Data availability.** The Gene Expression Omnibus (GEO) data referenced during the study are available in a public repository from the GEO website (https://www.ncbi.nlm.nih.gov/geo/). The authors declare that all the other data supporting the findings of this study are available within the article and its Supplementary Information files and from the corresponding author on reasonable request.

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

## Acknowledgements

We thank the Institutional Development Award (IDeA) grant from the National Institute of General Medical Sciences of the National Institutes of Health under grant number P20 GM103639 for the use of Histology and Immunohistochemistry Core, which provided immunohistochemistry and image analysis service. We thank Prof Muniswamy Madesh for the generous gift of GCaMP2-mt construct. This work was supported by National Institutes of Health Grant NHLBI HL120585 and CA136494 (to P.M.), CA157481 (to R.B.) and AR064211 (to L.T.).

## Author contributions

Conception and design: P.K.C., R.B. and P.M. Development of methodology: P.K.C., S.B.M., X.X., S.K.D.D., S.S. and M.Z. Acquisition of data: P.K.C., S.B.M., S.S., M.J., D.Y., R.Z., K.D., S.M., V.N. and S.K.D.D. Analysis and interpretation of data: P.K.C., S.B.M., S.S., M.J., D.Y., R.Z., K.D., S.M., V.N., L.T. and P.M. Writing, review and/or revision of the manuscript: P.K.C., D.D., D.Y., R.M., K.M., R.B. and P.M. Administrative, technical or material support: R.M., K.M., S.M., R.B. and P.M. Study supervision: P.M.

## Additional information

**Competing financial interests:** The authors declare no competing financial interests.

