## [Peer Review File · Nature Communications]

Reviewers' comments:

Reviewer #1 (Mitochondria metabolism in cancer)
(Remarks to the Author):

The contribution by Chakraborty and colleagues is an original and interesting piece of work suggesting that dysregulation of mitochondrial Ca²⁺ homeostasis by the overexpression of the negative regulator of the mitochondrial Ca²⁺ uniporter MICU1 in ovarian carcinomas is responsible for diverting pyruvate from the action of pyruvate dehydrogenase (PDH) triggering in this way the upregulation of glycolysis and cellular chemoresistance to cisplatin. In other words, emphasizes another mechanism by which pyruvate dehydrogenase kinase (PDK) could be playing a role in metabolic rewiring in cancer cells (see also McFate et al., JBC 283, 22700 (2008), Pate et al., EMBO J. 33, 1454 (2014)). Cell-death resistance to cisplatin-induced Ca²⁺ release from intracellular stores is ascribed to the protection exerted by MICU1 for mitochondrial Ca²⁺ entry and the subsequent unloading of the apoptotic response. In addition, and consistent with the work presented, they provide relevant information regarding the clinical outcome for ovarian cancer patients with overexpression of MICU1 mRNA and protein and the expression of the inactive (phosphorylated) PDH. Moreover, they show that tumor xenographs in nude mice bearing silenced MICU1 are much less aggressive than tumors derived from the parental cells. The work, for the most part of it, is clearly presented and documented and accompanied by appropriate controls. However, the paper fails to demonstrate the role of MICU1 expression level in controlling intramitochondrial Ca²⁺ concentrations.

MAJOR POINTS:

1.- Intramitochondrial Ca²⁺ levels need to be documented in response to MICU1 regulated expression with appropriate Ca²⁺ sensitive proteins targeted to mitochondria because a major claim of the paper is based on the role played by intramitochondrial Ca²⁺ in metabolic dysregulation. Nowadays, Rhod-2 fluorescence is not reliable enough as a mitochondrial Ca²⁺ sensitive probe.

2.- The observed changes in maximum respiratory rate (FCCP stimulated) in response to the regulated expression of MICU1 need some explanation and additional investigation. Is it possible that MICU1 could be primarily affecting the expression/assembly of respiratory complexes (RC)? Analysis of RC in BN-gels could help in this regard.

Minor points:

1.-On page 6, it is mentioned that mitochondrial copy number remains unaltered. The authors have determined the expression of a mitochondrial gene and assumed that the copy number of mtDNA is not affected. This should be clarified in the text or alternatively the abundance of a mitochondrial gene relative to the abundance of a nuclear one should be determined.

2.- On page 8, the Ferrick reference has not been converted. The two c's of Cytochrome C should be in lower case.

3.- On page 10, the abbreviation Pyr6 should be spelled out.

4.- The effect of FCCP on the TMRM Fluorescence Intensity should be incorporated in Figs. 3A,B. In addition, since in the Y-axis it is being represented the Relative Fluorescence Intensity the scale can be simplified representing data in the 0 to 20 range.

5.- The claimed association/dissociation of pyruvate dehydrogenase phosphatase (PDP) with PDH in response to the silencing/overexpression of MICU1 (Fig. 6) needs quantification and statistical evaluation.

Reviewer #2 (Calcium metabolism in Cancer)
(Remarks to the Author):

The paper of Chakraborty et al. describes new mechanism of chemoresistance in ovarian cancer cells, it demonstrates that MICU1 drives aerobic glycolysis in ovarian cancer by modulating the pyruvate dehydrogenase (PDH). The authors suggest that MICU1 could serve as a novel therapeutic target to normalize metabolic aberrations. Although the results are interesting and the main message really important, I have a number of major points to be clarified before the paper becomes acceptable for Nature Communications.

Major points

- Most of the experiments described in this manuscript were performed in CP20 and OV90 cells. It is however unclear as to why the authors chose these cell lines among all those they tested. Their choice should be explained.
- Throughout the manuscript, the authors have used only one test for proliferation/migration/invasion. Confirmation of these preliminary results with different assays would go a long way to strengthen the authors' conclusions.
- The authors chose cisplatin as their reference chemotherapeutic drug. Are similar results obtained with a different drug, or are they exclusive to this compound?
- In Fig. 3F, the authors use mitoTEMPO to scavenge ROS. A control experiment showing that this is actually the case in their models is missing. They could use MitoSOX dye, as shown in Fig. 3E.
- Fig. 4A: while the spike in Ca^{2+}_m induced by cisplatin seems indeed different between siCTL and siMICU1-treated cell, this difference disappears within 400s. Are the authors suggesting that cell fate will be drastically altered by these initial 400s? They should carefully study the kinetics of their responses, and show results proving long-lasting differences between their conditions.
- Fig. 4E: "...indicating absence of Ca^{2+} release from ER without cisplatin treatment." This point is very important, and need to be clearly demonstrated. Are ER Ca^{2+} stores really identical in shRNA-treated cells? The authors should measure them in their models, using ionomycin in Fura2-loaded cells, or any other method available to them.
- page 12: they authors hypothesize that MICU1 will lead to " Ca^{2+} overload in the mitochondria". They should show evidence of this: dyes are available for that kind of measurement, and they need to substantiate the fact that calcium homeostasis is indeed modified by MICU1 silencing.
- page 13: "a direct consequence of MICU1 driven lowering of Ca^{2+}_m ". Could the authors indicate where they have shown that MICU1 overexpression decreases Ca^{2+}_m ? Does not this statement contradict results of Fig. 4D?
- The in-vivo results are quite puzzling: first, Fig. 7B seems to indicate that all 10 animals injected with shCTL cells died on the same day. Is it really that synchronized? Second, the conclusion that cells without MICU1 are more sensitive to cisplatin is invalidated by these experiments. Indeed, on both populations (shCTL and shMICU1 injected mice), cisplatin treatment gave the same extra 7 days of survival. The conclusion from this experiment should rather be that MICU1 and cisplatin act through distinct pathways, as also illustrated in Fig. 7A where tumor growth inhibition by cisplatin seems to be proportionally identical between shCTL and shMICU1.

Reviewer #3 (mechanisms of chemotherapy resistance)
(Remarks to the Author):

The MS by Chakraborty and colleagues is a follow up of the authors' early observation that small interfering RNA-mediated silencing of the mitochondrial uniporter MICU1 is able to sensitize ovarian cancer cells to positively charged gold nanoparticles. This observation suggested a potential role of MICU1 in drug resistance.

It is known that MICU1 resides in the inner mitochondrial membrane (IMM) and required for Ca^{2+} uptake through mitochondrial Ca^{2+} 'uniporter' (MCU), a calcium-selective ion channel responsible for low-affinity calcium uptake into the mitochondrial matrix. It has been shown that MCU expression is correlated with cancer progression in sporadic studies (prostate, colon and breast). However, the functional role for MICU1 in cancer is still unknown. In the present MS the authors suggesting new function of MICU1 that regulates metabolic fate and confers chemoresistance in ovarian cancer. Moreover, they tried to make a link between obtained observation and cancer prognosis. Although the message of this MS is interesting and the quality of submitted MS is high, presented data raise several questions that the authors have to address.

1. Work from Rizzuto's lab revealed that MICU1 and MICU2 finely tune the mitochondrial Ca^{2+} uniporter by exerting opposite effects on MCU activity. It is also known that MICU1 and MICU2, which, both in purified lipid bilayers and in intact cells, stimulate and inhibit MCU activity, respectively. Keeping that in mind inhibition of MICU1 should even stronger influence MCU activity. Downregulation of MICU1 does not influence expression level of MCU and EMRE, but MICU2 showed modest decrement. How MICU2 should act in these conditions? What is the level of expression of MICU2 in ovarian cancers?

2. The authors suggested that removal of MICU1 leads to mitochondrial calcium accumulation. If MICU2 is acting properly, then the balance in calcium transport that regulated by MICU1 and MICU2 is disturbing. Data presented in Fig. 4a should be explained, keeping in mind even small changes in expression of MICU2. Moreover, removal of MICU1 on OV90 cells did not changed significantly level of calcium. It is important to note that cisplatin causes similar release of $[\text{Ca}^{2+}]_{\text{ER}}$ in presence or absence of MICU1 in OV90 cells (fig. 4), but appearance of calcium in cytosol is dangerous. This should be explained and stronger evidence that MICU1 restricts chemotherapy-stimulated entry of Ca^{2+} into the mitochondria, even when large concentration of Ca^{2+} is released from ER, thus conferring chemoresistance of ovarian cells should be presented.

3. Indeed, the silencing of MICU1 influenced anchorage-independent clonal growth. However, in two out of three significant decrease (up to 50%) in colony formation was observed even after transfection of control siRNAs (see Fig. 2 a-c), which questioning the response of cells to more specific silencing of MICU1. Moreover, as the authors stated the role of calcium on clonal growth and differentiation has been well established in human bronchial, leukemic and epidermal cells. Several calcium channels are involved in this process. How elimination of one protein that involves in the regulation of function of just one channel can so drastically influence the process of clonal growth? This should be explained and discussed more carefully.

4. The authors suggested that experiments with short incubation with cisplatin mimic clinical cisplatin therapy schedule, which is not correct. To draw this conclusion the authors should show pumping out of cisplatin from the cells, the effect that can simply be observed in clinical settings.

5. Cisplatin-induced apoptosis is not always perpetuates with drop of MMP. Release of cytochrome c and other intermembrane-located proteins might occur via formation of specific Bax/Bak-induced pores that lead to activation of caspase-9, etc, the process independent on Ca^{2+} . Indeed, upon treatment with cisplatin caspase-9 is activating in cells expressing MICU1 (Fig. 3c) and removal of this protein is just accelerate this process. It is unclear why in cells with active caspase-9 PARP is not cleaved? It also raises the question concerning the role of calcium in this process. If removal of

MICU1 just accelerates cisplatin-induced death, then the role of MICU1 in regulation of sensitivity to treatment is not the primary.

6. The authors concluded that expression level of PDK and pPDH in ovarian cancers is higher than in normal cells. However, data from Fig 6a contradict it. Level of PDK in OSE cells is similar to OVCAR2 and OV90 cells. On the other hand, level of pPDH is not correlated with level of PDK. This discrepancy should be explained.

7. Unfortunately, the total level of PDH in cells with siMICU1 is increasing (see Fig. 6C). In this case decreasing of the level of pPDH should be explained.

8. Why shCTL-implantation group formed statistically significant larger tumors?

9. Data in Fig. 7a showed that treatment with cisplatin is equally efficient as removal of MICU1, which again raises the questions whether this removal sensitizes cells to treatment?

10. What happened with α -ketoglutarate- and isocitrate-dehydrogenases in cells deficient in MICU1?

11. Discussion is too long and unclear.

Response to the reviewers' comments

Response: At the outset we would like to mention that we appreciate the editor for giving us the opportunity to revise the paper. We took the reviewers' recommendations to heart and worked diligently to address all of the concerns raised by performing additional experiments, providing new data and making appropriate changes in the manuscript.

Below is our point-by-point response to the reviewers' concerns:

Reviewer#1:

The contribution by Chakraborty and colleagues is an original and interesting piece of work suggesting that dysregulation of mitochondrial Ca²⁺ homeostasis by the overexpression of the negative regulator of the mitochondrial Ca²⁺ uniporter MICU1 in ovarian carcinomas is responsible for diverting pyruvate from the action of pyruvate dehydrogenase (PDH) triggering in this way the upregulation of glycolysis and cellular chemoresistance to cisplatin. In other words, emphasizes another mechanism by which pyruvate dehydrogenase kinase (PDK) could be playing a role in metabolic rewiring in cancer cells (see also McFate et al., JBC 283, 22700 (2008), Pate et al., EMBO J. 33, 1454 (2014)). Cell-death resistance to cisplatin-induced Ca²⁺ release from intracellular stores is ascribed to the protection exerted by MICU1 for mitochondrial Ca²⁺ entry and the subsequent unslashing of the apoptotic response. In addition, and consistent with the work presented, they provide relevant information regarding the clinical outcome for ovarian cancer patients with overexpression of MICU1 mRNA and protein and the expression of the inactive (phosphorylated) PDH. Moreover, they show that tumor xenografts in nude mice bearing silenced MICU1 are much less aggressive than tumors derived from the parental cells. The work, for the most part of it, is clearly presented and documented and accompanied by appropriate controls. However, the paper fails to demonstrate the role of MICU1 expression level in controlling intramitochondrial Ca²⁺ concentrations.

Response: We appreciate the reviewer's encouraging comments and thoughtful suggestions. In agreement with the reviewer, we performed additional experiments to demonstrate modulation of intramitochondrial Ca²⁺ by MICU1 and its role in glycolysis and chemoresistance. Briefly, in addition to chemical probes, as suggested by the reviewers, we have now used protein based mitochondrial Ca²⁺ indicator GCaMP2-mt to determine the regulation of mitochondrial Ca²⁺ ([Ca²⁺]_m) by MICU1 and its role in drug sensitivity in ovarian cancer models (**Figures 4D, F, Supporting Figures S10, S11**). Further to investigate a cause for increased oxygen consumption rate (OCR) in MICU1 silenced cells, we determined the protein expression levels of the key ETC components. . However, we did not observe any appreciable changes in the expression levels of the ETC components (**Supporting Figure S15**). We next investigated individual mitochondrial complex activity from MICU1 silenced cells and observed a significant increase in Complex III activity that is in agreement with the observed increase in mitochondrial ROS (mtROS) (**Supporting Figure S16**). Future investigations will be focused on how MICU1 regulates Complex III activity. Furthermore, as suggested by the reviewer, we have now

quantified the immunoblots demonstrating PDH-PDP interactions using NIH Image J and presented as a bar graph (**Figures 6 C, E, H, K**). We have also corrected all the minor issues as suggested by the reviewer. Please find our point-by-point response to the reviewer's comments in the following section.

Reviewer 1: MAJOR POINTS:

Reviewer 1: 1. *Intramitochondrial Ca²⁺ levels need to be documented in response to MICU1 regulated expression with appropriate Ca²⁺ sensitive proteins targeted to mitochondria because a major claim of the paper is based on the role played by intramitochondrial Ca²⁺ in metabolic dysregulation. Nowadays, Rhod-2 fluorescence is not reliable enough as a mitochondrial Ca²⁺ sensitive probe.*

Response: We appreciate the reviewer's thoughtful suggestions. As suggested we performed our [Ca²⁺]_m measurements using the highly sensitive and selective protein-based mitochondrial Ca²⁺ probe GCaMP2-mt [Kirichok et al., *Nature* 427, 360-364 (2004) and Doonan et al., *The FASEB Journal* vol. 28 no. 11, 4936-4949 (2014)]. We have repeated our experiments in different cell lines expressing GCaMP2-mt to further confirm regulation of [Ca²⁺]_m by MICU1 and its effect on cisplatin treatment (**Figures 4D, F, Supporting Figures S10, S11**). In brief, stable cell lines shCTL-OV90 and shMICU1-OV90, were transfected with GCaMP2-mt and mitochondrial Ca²⁺ response monitored after cisplatin exposure (**Figure 4D**). Compared to scrambled control, MICU1 knockdown increased [Ca²⁺]_m by ~5 fold upon cisplatin treatment. Similar effects were observed in siMICU1-CP20 cells compared to siCTL-CP20 cells (**Supporting Figure S10**). Ectopic expression of MICU1 in non-malignant FTE-188 cells having low endogenous MICU1 levels, inhibited cisplatin induced increase in [Ca²⁺]_m (**Figure 4F**). Taken together, these additional experiments further confirm our hypothesis that MICU1 functions as a gatekeeper preventing [Ca²⁺]_m overload in response to cisplatin thereby conferring resistance to cisplatin. In accordance, two recent reports demonstrated that MICU1 serves as a molecular gate keeper preventing [Ca²⁺]_m overload, increasing survival in post-natal life [Antony et al., *Nature Communications* 7:10955 (2016) and Liu et al., *Cell Reports* 16(6), 1561-73 (2016)]

Reviewer 1: 2. *The observed changes in maximum respiratory rate (FCCP stimulated) in response to the regulated expression of MICU1 need some explanation and additional investigation. Is it possible that MICU1 could be primarily affecting the expression/assembly of respiratory complexes (RC)? Analysis of RC in BN-gels could help in this regard.*

Response: We appreciate the reviewer's thoughtful suggestions. As suggested we have performed immunoblotting for the components of the mitochondrial complexes. Using commercially available kits, we performed these experiments from isolated mitochondria of both OV90 and CP20 cells with or without MICU1 silencing. We did not observe any significant differences in the protein expression

of the components of the respiratory complexes (**Supporting Figure S15**). Next, measuring individual mitochondrial Complex activity, we observed that silencing MICU1 in CP20 or OV90 cells significantly increased Complex III activity (**Supporting Figure S16**), while those for Complex I, II and IV remained unchanged (**Supporting Figure S16**). Complex III activity is associated with generation of mitochondrial ROS (mtROS) [Bleier et al., Biochimica et Biophysica Acta (BBA) - Bioenergetics Volume 1827, Issues 11–12 (2013)] and corroborates our results demonstrating higher mtROS in MICU1 silenced cells. Since higher ROS levels render cells vulnerable to cytotoxic stress [Fulda et al., International Journal of Cell Biology Volume 2010, 214074 (2010)], a similar mechanism could be envisioned for MICU1 silenced cells being sensitized to cisplatin. Future investigations will be focused on how MICU1 regulates Complex III activity. We have incorporated these findings in the results and discussion section of the revised manuscript.

Reviewer 1: Minor points:

Reviewer 1:1. *On page 6, it is mentioned that mitochondrial copy number remains unaltered. The authors have determined the expression of a mitochondrial gene and assumed that the copy number of mtDNA is not affected. This should be clarified in the text or alternatively the abundance of a mitochondrial gene relative to the abundance of a nuclear one should be determined.*

Response: We appreciate the reviewer's suggestions. We have now repeated this experiment using real-time PCR and normalized the abundance of mitochondrial gene ND1 relative to the abundance of the nuclear gene 3692 (RPLP0) and replaced the previous supporting Figure S3 with the newer version. Our conclusions remain unchanged in that there were no significant differences in relative mRNA expression between siMICU1 or shMICU1 cells compared to their respective controls. We have revised the results and methods section with this new information.

Reviewer 1:2. *On page 8, the Ferrick reference has not been converted. The two c's of Cytochrome C should be in lower case.*

Response: We regret these mistakes. We have now corrected these errors in the revised manuscript.

Reviewer1: 3. *On page 10, the abbreviation Pyr6 should be spelled out.*

Response: As suggested by the reviewer we have now expanded the abbreviation for Pyr6 as (N-[4-[3,5-Bis(trifluoromethyl)-1H-pyrazol-1-yl]phenyl]-3-fluoro-4-pyridinecarboxamide)].

Reviewer 1: 4. *The effect of FCCP on the TMRM Fluorescence Intensity should be incorporated in Figs. 3A,B. In addition, since in the Y-axis it is being represented the Relative Fluorescence Intensity the scale can be simplified representing data in the 0 to 20 range.*

Response: We appreciate the reviewer's suggestions. As suggested we modified figures 3A and 3B and included FCCP control by repeating the experiments. Furthermore, as suggested, we have simplified the Relative Fluorescence Intensity scale representing data within the 0 to 20 range.

Reviewer1: 5. *The claimed association/dissociation of pyruvate dehydrogenase phosphatase (PDP) with PDH in response to the silencing/overexpression of MICU1 (Fig. 6) needs quantification and statistical evaluation.*

Response: We appreciate the reviewer's suggestions. As suggested we have quantified the immunoblots (from three independent experiments) in Figure 6 pertaining to the association/dissociation of pyruvate dehydrogenase phosphatase (PDP) with PDH in response to the silencing/overexpression of MICU1, using NIH-Image J and graphically represented as mean \pm SEM. The revised and re-interpreted results based on the quantitative analysis are now incorporated in Figures 6 C, E, H, K.

Reviewer#2: The paper of Chakraborty et al. describes new mechanism of chemoresistance in ovarian cancer cells, it demonstrates that MICU1 drives aerobic glycolysis in ovarian cancer by modulating the pyruvate dehydrogenase (PDH). The authors suggest that MICU1 could serve as a novel therapeutic target to normalize metabolic aberrations. Although the results are interesting and the main message really important, I have a number of major points to be clarified before the paper becomes acceptable for Nature Communications.

Response: We appreciate the reviewer's encouraging remarks and thoughtful suggestions. We have revised the manuscript as suggested by the reviewer. Please find our point-by-point response to the reviewer's comments below;

Reviewer 2: 1. *Most of the experiments described in this manuscript were performed in CP20 and OV90 cells. It is however unclear as to why the authors chose these cell lines among all those they tested. Their choice should be explained.*

Response: We appreciate the reviewer's thoughtful suggestions. Though the expression of MICU1 was significantly higher in most of the ovarian cancer cell line, we selected CP20 and OV90 for our

studies because we wanted to examine the role of MICU1 in chemoresistant and aggressive cancer cell lines. The CP20 cell line was developed by treating cells with increasing concentrations of cisplatin and presents a chemoresistance cell line model [Sood AK et al., *The American journal of pathology* **158**, 1279-1288 (2001)]. The OV90 cell line was derived from ascites of a grade 3, stage IIIc ovarian cancer patient and hence provides a relevant model to demonstrate proof-of-concept role of MICU1 in ovarian cancer [Provencher DM et al., *In vitro cellular & developmental biology. Animal* **36**, 357-361 (2000)]. The rationale has now been explained in the result section.

Reviewer2: 2. Throughout the manuscript, the authors have used only one test for proliferation/migration/invasion. Confirmation of these preliminary results with different assays would go a long way to strengthen the authors' conclusions.

Response: We appreciate the reviewer's suggestions. As suggested, we have now expanded our studies with additional tests for proliferation (BrDU incorporation assay), migration/invasion (Boyden Chamber based Calcein-AM assay and gelatin degradation assay). Similar to the CyQUANT assay, the BrDU incorporation assay also demonstrated a dose dependent reduction in BrDU incorporation upon cisplatin treatment in the MICU1 silenced cells, compared to the control, indicating sensitization of tumor cells to cisplatin upon MICU1 silencing. Furthermore, similar to the CyQUANT assay, MICU1 silencing alone did not inhibit proliferation as evidenced by the absence of changes in BrDU incorporation upon MICU1 silencing. (**Supporting Figures S4 A,B**) We next performed the Boyden-Chamber assay and quantified cell migration or invasion by determining Calcein-AM fluorescence to further support a role of MICU1 in cell migration and invasion (**Figures 2D,E**). Silencing MICU1 in CP20 or OV90 cells significantly reduced migration (~72% in CP20 and 63% in OV90 cells) and invasion (~76% in CP20 and 74% in OV90 cells). In addition, we performed the gelatin-degradation assay to further support a role for MICU1 in cellular invasion (**Figures 2F,G**). Extent of degradation of the gelatin matrix provides a measure of the invasive potential of cancer cells and our results demonstrate that MICU1 silenced CP20 or OV90 cells have significantly compromised gelatin-degrading capabilities. These new experiments and the results have been now been incorporated in the methods, results and discussion sections of the revised manuscript.

Reviewer2: 3. The authors chose cisplatin as their reference chemotherapeutic drug. Are similar results obtained with a different drug, or are they exclusive to this compound?

Response: We appreciate the reviewer's insightful comments. We have now investigated the effect of other standard chemotherapeutics such as topotecan, paclitaxel and doxorubicin on ovarian cancer cell proliferation upon MICU1 silencing. We observed that MICU1 silencing sensitized ovarian cancer cell to topotecan, paclitaxel as well as doxorubicin (**Supporting Figures S5**). Interestingly, prior reports suggest that doxorubicin [*Biochim Biophys Acta*. 1813(6):1144-52.(2011)] and paclitaxel [Kidd et al., *The Journal of Biological Chemistry* 277, 6504-6510 (2002)] affect calcium homeostasis in

cancer cells to exert their cytotoxic effects and further support our hypothesis that MICU1 imparts resistance to their cytotoxicity by serving as a gate keeper preventing $[Ca^{2+}]_m$ overload.

Reviewer2: 4. *In Fig. 3F, the authors use mitoTEMPO to scavenge ROS. A control experiment showing that this is actually the case in their models is missing. They could use MitoSOX dye, as shown in Fig. 3E.*

Response: We appreciate the reviewer's thoughtful suggestions. As suggested we performed control experiments showing mitoTEMPO mediated scavenging of mtROS as evident by reduced staining with the MitoSOX dye (**Supporting Figure S7**). Pre-incubation of the MICU1 silenced CP20 cells with the mitochondrial ROS scavenger mitoTEMPO (10 μ M) resulted in quenching of mtROS as determined by a reduction in MitoSOX staining compared to untreated control (**Supporting Figure S7**). Extending these investigations, the fold change in of apoptosis in cisplatin treated MICU1 silenced cells significantly decreased in the presence of mitoTEMPO (**Figure 3F**). These results suggest that silencing MICU1 generates mtROS in ovarian cancer cells facilitating a cisplatin-mediated apoptotic response.

Reviewer2: 5. *Fig. 4A: while the spike in Ca^{2+}_m induced by cisplatin seems indeed different between siCTL and siMICU1-treated cell, this difference disappears within 400s. Are the authors suggesting that cell fate will be drastically altered by these initial 400s? They should carefully study the kinetics of their responses, and show results proving long-lasting differences between their conditions.*

Response: We appreciate the reviewer's thoughtful comments for the short-lived changes in $[Ca^{2+}]_m$. As suggested we now tested the $[Ca^{2+}]_m$ dynamics using GCaMP2-mt construct. It was evident from the **Figures 4D, F** that cisplatin treatment resulted in $[Ca^{2+}]_m$ changes within 300-600s range. It is notable that MICU1 silencing has been demonstrated to exhibit changes in $[Ca^{2+}]_m$ within similar time frames [Mallilankaraman et al., *Cell* **151**, 630-644 (2012)]. We posit the rapid change in $[Ca^{2+}]_m$ results from influx of the cisplatin-induced release of Ca^{2+} from endoplasmic reticulum (ER), but nevertheless were sufficient enough to activate dehydrogenases and signaling cascades [Rizzuto et al., *Science*. 262:744–747 (1993) and Denton et al., *Biochim Biophys Acta* **1787**, 1309-1316 (2009)].

Reviewer2: 6. *Fig. 4E: "...indicating absence of Ca^{2+} release from ER without cisplatin treatment." This point is very important, and need to be clearly demonstrated. Are ER Ca^{2+} stores really identical in shRNA-treated cells? The authors should measure them in their models, using ionomycin in Fura2-loaded cells, or any other method available to them.*

Response: We appreciate the reviewer's thoughtful suggestions. We have now repeated the experiment as suggested. We utilized Thapsigargin (5 μ m) as a trigger for ER Ca^{2+} release and found

similar increase in cytosolic calcium ($[Ca^{2+}]_i$) in the presence/or absence of MICU1 as determined in OV90 cells loaded with cytosolic calcium indicator dye Fura-2. The results suggest that there is no significant difference in $[Ca^{2+}]_{ER}$ of either control or MICU1 silenced cells (**Supporting Figure S12**).

Reviewer2: 7. *page 12: they authors hypothesize that MICU1 will lead to "Ca2+ overload in the mitochondria". They should show evidence of this: dyes are available for that kind of measurement, and they need to substantiate the fact that calcium homeostasis is indeed modified by MICU1 silencing.*

Response: We appreciate the reviewer's thoughtful suggestions. As suggested we performed our $[Ca^{2+}]_m$ measurements using the highly sensitive and selective protein-based mitochondrial Ca^{2+} probe GCaMP2-mt [Kirichok et al., *Nature* 427, 360-364 (2004) and Doonan et al., *The FASEB Journal* vol. 28 no. 11, 4936-4949 (2014)]. We have repeated our experiments in different cell lines expressing GCaMP2-mt to further confirm regulation of $[Ca^{2+}]_m$ by MICU1 and its effect on cisplatin treatment (**Figures 4D, F, Supporting Figures S10, S11**). In brief, stable cell lines shCTL-OV90 and shMICU1-OV90, were transfected with GCaMP2-mt and mitochondrial Ca^{2+} response monitored after cisplatin exposure (**Figure 4D**). Compared to scrambled control, MICU1 knockdown increased $[Ca^{2+}]_m$ by ~5 fold upon cisplatin treatment. Similar effects were observed in siMICU1-CP20 cells compared to siCTL-CP20 cells (**Supporting Figure S10**). Ectopic expression of MICU1 in non-malignant FTE-188 cells having low endogenous MICU1 levels, inhibited cisplatin induced increase in $[Ca^{2+}]_m$ (**Figure 4F**). Taken together, these additional experiments further confirm our hypothesis that MICU1 functions as a gatekeeper preventing $[Ca^{2+}]_m$ overload in response to cisplatin thereby conferring resistance to cisplatin. In accordance, two recent reports demonstrated that MICU1 serves as a molecular gate keeper preventing $[Ca^{2+}]_m$ overload, increasing survival in post-natal life [Antony et al., *Nature Communications* 7:10955 (2016) and Liu et al., *Cell Reports* 16(6), 1561-73 (2016)]

Reviewer2: 8. *page 13: "a direct consequence of MICU1 driven lowering of Ca2+m". Could the authors indicate where they have shown that MICU1 overexpression decreases Ca2+m? Does not this statement contradict results of Fig. 4D?*

Response: We appreciate the reviewer's thoughtful comments and agree that in absence of cisplatin, only a marginal decrease in $[Ca^{2+}]_m$ in OSE cells expressing Flag-MICU1 is observed (**Figure 4E**). However, ectopic expression of Flag-MICU1 in OSE cells significantly prevented cisplatin-mediated $[Ca^{2+}]_m$ overload as compared to the EV-expressing control cells (**Figure 4E, F**), confirming the role of MICU1 as a negative regulator of mitochondrial calcium uptake. Importantly, OSE cells harboring Flag-MICU1 demonstrate higher pPDH (ser293) expression (**Supporting Figure S17**) and decrease in PDH activity as compared to the control cells (**Figure 6I**). These results further support that though ectopic expression of MICU1 in absence of any stimulation decreases $[Ca^{2+}]_m$ marginally, but this small decrement is significant enough to inhibit PDP-PDH interaction (**Figure 6G**) leading to

pPDH accumulation. Indeed MICU1 silencing alone in HeLa cells did not affect the $[Ca^{2+}]_m$ of unstimulated cells yet possessed significantly different phosphoPDH levels [Mallilankaraman et al., *Cell* **151**, 630-644 (2012)].

Reviewer2: 9. *The in-vivo results are quite puzzling: first, Fig. 7B seems to indicate that all 10 animals injected with shCTL cells died on the same day. Is it really that synchronized? Second, the conclusion that cells without MICU1 are more sensitive to cisplatin is invalidated by these experiments. Indeed, on both populations (shCTL and shMICU1 injected mice), cisplatin treatment gave the same extra 7 days of survival. The conclusion from this experiment should rather be that MICU1 and cisplatin act through distinct pathways, as also illustrated in Fig. 7A where tumor growth inhibition by cisplatin seems to be proportionally identical between shCTL and shMICU1.*

Response: We appreciate the reviewer's concern and regret prior lack of clarification. Before the initiation of treatment, the animals were randomized in a blinded fashion. We observed that all the animals injected with shCTL cells (tumor growth study as well as survival study) developed large tumors (~1.469g), were moribund, appeared in distress and hence were euthanized along with the experimental groups for tumor growth study on the same day to avoid undue suffering according to the animal protocol approved by the institutional Animal Care and Use Committee (IACUC). We used same IACUC approved criteria to euthanize animals for all other groups in survival studies. We have now corrected and included the median survival values for the different groups (**Figure 7B**). It is evident from the table that cisplatin treatment in shCTL group yielded 2 extra days of survival while animals in shMICU1 group survived for 43 days. In addition, cisplatin therapy further prolonged their survival by 10 more days. It is also important to note that though tumor growth inhibition by cisplatin appears to be proportionally identical between shCTL and shMICU1 groups; in the shMICU1 group, 4 out of 5 animals showed negligible tumor formation upon cisplatin treatment (80% of the sample size) as compared to 1 out of 5 animals in shCTL+cisplatin group (20% of the sample size), supporting a role of MICU1 in drug resistance. However, we agree that there may be other unknown pro-oncogenic function of MICU1 that will evolve over time.

Reviewer3: The MS by Chakraborty and colleagues is a follow up of the authors' early observation that small interfering RNA-mediated silencing of the mitochondrial uniporter MICU1 is able to sensitize ovarian cancer cells to positively charged gold nanoparticles. This observation suggested a potential role of MICU1 in drug resistance. It is known that MICU1 resides in the inner mitochondrial membrane (IMM) and required for Ca^{2+} uptake through mitochondrial Ca^{2+} 'uniporter' (MCU), a calcium-selective ion channel responsible for low-affinity calcium uptake into the mitochondrial matrix. It has been shown that MCU expression is correlated with cancer progression in sporadic studies (prostate, colon and breast). However, the functional role for MICU1 in cancer is still unknown. In the present MS the authors suggesting new function of MICU1 that regulates metabolic fate and confers chemoresistance in ovarian cancer. Moreover, they tried to make a link between obtained observation and cancer

prognosis. Although the message of this MS is interesting and the quality of submitted MS is high, presented data raise several questions that the authors have to address.

Response: We appreciate the reviewer's appreciation and thoughtful comments about our work. We have revised the manuscript as suggested by performing additional experiments and incorporating new data that has further improved the quality of the manuscript. Briefly, we have further investigated any role of MICU2 in chemosensitization of ovarian cancer cells. However, we did not find any role of MICU2 in cisplatin sensitivity. Additionally, MICU2 silencing alone or in a MICU1 ablated background resulted in a drop in $[Ca^{2+}]_m$ upon cisplatin treatment, indicating antagonistic roles of MICU1 and MICU2. Furthermore, we have also demonstrated the roles of cisplatin mediated $[Ca^{2+}]_{ER}$ release and subsequent $[Ca^{2+}]_m$ overload as the causative cytotoxic effect in MICU1 silenced cells. Please find our point-by-point response to the reviewer's comments in the following section.

Reviewer3: 1. *Work from Rizzuto's lab revealed that MICU1 and MICU2 finely tune the mitochondrial Ca^{2+} uniporter by exerting opposite effects on MCU activity. It is also known that MICU1 and MICU2, which, both in purified lipid bilayers and in intact cells, stimulate and inhibit MCU activity, respectively. Keeping that in mind inhibition of MICU1 should even stronger influence MCU activity. Downregulation of MICU1 does not influence expression level of MCU and EMRE, but MICU2 showed modest decrement. How MICU2 should act in these conditions? What is the level of expression of MICU2 in ovarian cancers?*

Response: We appreciate the reviewer's thoughtful comments. Indeed MICU1 silencing reduces MICU2 expression levels that might synergize the changes in $[Ca^{2+}]_m$. To address this concern first we tested expression of MICU2 in a panel of ovarian cancer cell lines and non-malignant normal ovarian cell lines. Compared to non-malignant FTE-188 and OSE control cells, most of the ovarian cancer cell lines expressed higher levels of MICU2 (**Supporting Figure S6A**). To investigate any role of MICU2 in drug resistance, we further expanded our study by silencing MICU2 (**Supporting Figure S6B**) in CP20 and OV90 cells and probing the effects on cisplatin sensitization. Unlike MICU1, MICU2 silencing did not enhance cisplatin activity, suggesting MICU2 is not a significant contributor of chemoresistance in ovarian cancer cells (**Supporting Figures S6D, E**).

Reviewer3: 2. *The authors suggested that removal of MICU1 leads to mitochondrial calcium accumulation. If MICU2 is acting properly, then the balance in calcium transport that regulated by MICU1 and MICU2 is disturbing. Data presented in Fig. 4a should be explained, keeping in mind even small changes in expression of MICU2. Moreover, removal of MICU1 on OV90 cells did not changed significantly level of calcium. It is important to note that cisplatin causes similar release of $[Ca^{2+}]_{ER}$ in presence or absence of MICU1 in OV90 cells (fig. 4), but appearance of calcium in cytosol is dangerous. This should be explained and stronger evidence that MICU1 restricts chemotherapy-stimulated entry of Ca^{2+} into the mitochondria, even when large concentration of Ca^{2+} is released from ER, thus conferring chemoresistance of ovarian cells should be presented.*

Response: We appreciate the reviewer's thoughtful comments. We first determined the expression of MICU2 in a panel of OvCa cell lines and found MICU2 expression to be moderately higher in most of the OvCa cells as compared to non-malignant OSE and FTE-188 cells (**Supporting Figure S6A**). Silencing of MICU2 was achieved by transient transfection with siRNA (**Supporting Figure S6B**). However, silencing of MICU2 did not affect the expression of MICU1 (**Supporting Figure S6B**), but silencing of MICU1 caused modest decrease in MICU2 expression (**Supporting Figure S6C**). These findings are in agreement with recently reported studies [Patron *et al.*, *Mol Cell.* 53 (5): 726–737 (2014)]. We further investigated any possible roles of MICU2 in OvCa chemoresistance. We determined the effects of MICU2 silencing towards cisplatin sensitization in CP20 and OV90 cells. The results clearly demonstrated that silencing MICU2 did not enhance cisplatin activity in CP20 and OV90 cells (**Supporting Figure S6D, E**). Next we investigated the changes in $[Ca^{2+}]_m$ upon MICU2 silencing in shCTL and shMICU1 cells in response to cisplatin. Loss of MICU2 resulted in lowering of $[Ca^{2+}]_m$ upon cisplatin stimulation in both the cells (**Supporting Figure S11**), thereby suggesting opposing roles of MICU1 and MICU2 towards cisplatin induced changes in $[Ca^{2+}]_m$. We have incorporated this finding in the revised manuscript.

Further to evaluate the importance of cisplatin induced $[Ca^{2+}]_{ER}$ release in imparting cellular stress, we attempted to mitigate the $[Ca^{2+}]_{ER}$ release by pre-incubating cells with Pyr6 [(N-[4-[3,5-Bis(trifluoromethyl)-1H-pyrazol-1-yl]phenyl]-3-fluoro-4-pyridinecarboxamide) a selective inhibitor for $[Ca^{2+}]_{ER}$ release] [Scheilfer *et al.*, *Br J Pharmacol.* 167(8):1712-22 (2012)] prior to cisplatin treatment. Inhibition of ER calcium release by Pyr6 abrogated cisplatin induced changes in $[Ca^{2+}]_m$ in MICU1 silenced OvCa cells (**Figure 4H**). Additionally, the cytotoxic effects of cisplatin were minimized in Pyr6 pretreated siMICU1-CP20 cells, suggesting a requirement for ER calcium release by cisplatin, to execute anti-proliferative effects in MICU1 silenced cells (**Supporting Figure S12**). To further support our hypothesis that calcium efflux from the ER plays a significant role in the regulation of cell proliferation triggered by cisplatin, we treated ovarian cancer cell lines with a combination of cisplatin and/or the calcium chelating agent bis-(o-aminophenoxy) ethane-N,N,N',N'-tetra-acetic acid acetoxymethyl ester (BAPTA/AM). In the absence of BAPTA-AM, MICU1 silenced cells showed enhanced inhibition of cell proliferation by cisplatin. However, pre-treatment with BAPTA/AM abolished the effect of cisplatin on MICU1 silenced cells (**Supporting Figure S14**). These results further support our hypothesis that the inhibition of cisplatin induced ER-released Ca^{2+} uptake in the mitochondria by MICU1 confers cisplatin resistance in ovarian cells. Our hypothesis is further supported by two recent reports that demonstrated that the inability of ER Ca^{2+} to enter the mitochondria upon cisplatin treatment might cause chemoresistance in cancer cells [Liwei Ma *et al.*, *Aging Dis.* 2016 7(3): 254–266 (2016) and Shen *et al.*, *Oncol Lett.* 11(4): 2411–2419 (2016)].

Reviewer3: 3. *Indeed, the silencing of MICU1 influenced anchorage-independent clonal growth. However, in two out of three significant decrease (up to 50%) in colony formation was observed even after transfection of control siRNAs (see Fig. 2 a-c), which questioning the response of cells to more specific silencing of MICU1. Moreover, as the authors stated the role of calcium on clonal growth and differentiation has been well established in human bronchial, leukemic and epidermal cells. Several calcium channels are involved in this process. How elimination of one protein that involves in the*

regulation of function of just one channel can so drastically influence the process of clonal growth? This should be explained and discussed more carefully.

Response: We appreciate the reviewer's concern and regret for not clarifying the results properly. The inherent colony forming ability of different cell types are different. It is evident from the **Figure 2A** that inherent colony forming ability of control CP20 (siCTL) cell is different than control OV90 (siCTL), whereas there is no significant difference between siCTL- and shCTL- OV90 cells. The comparisons and quantifications were made between these individual controls and their MICU1 silenced counterparts, which exhibited significant reduction in colony forming abilities.

We appreciate the reviewer's thoughtful suggestions on the role of calcium channels in clonal growth and differentiation. The importance of calcium channels in cancer progression is well documented and each of them plays indispensable role(s) in this context. For example silencing of ORAI3A calcium channel caused robust inhibition of MCF-7 breast cancer cells through G₁ arrest [Faouzi et al., *J Membr Biol* 234, 47-56 (2010)]. Similarly, the silencing of the calcium channel TRPM8 resulted in drastic inhibition of the migration of PC3 prostate cancer cells, thus the role(s) of calcium channels extends beyond just inducers of cancer cell death [Gkika et al., *Oncogene* 29, 4611–4616 (2010)]. Finally, we would like to refer to the specific role of MICU1 (also known as CBARA1) in stem cell maintenance and clonal growth as described by Chen et al. [PLoS One. 8;8(5):e63653 (2013)] and their results suggest that CBARA1 is a marker for undifferentiated hESCs that plays a role in maintaining stemness, cell cycle progression, and proliferation. All of the above reports further support our finding that that MICU1 plays critical role in maintaining clonal growth in ovarian cancer cells.

Reviewer3: 4. The authors suggested that experiments with short incubation with cisplatin mimic clinical cisplatin therapy schedule, which is not correct. To draw this conclusion the authors should show pumping out of cisplatin from the cells, the effect that can simply be observed in clinical settings.

Response: We appreciate the reviewer's suggestion and agree with the comment. Since the focus of this paper is to investigate how MICU1 regulates glycolysis that confers chemoresistance, the short-term cisplatin incubation experiment is not providing any additional mechanistic insight to understand roles of MICU1 in glycolysis and chemoresistance. Therefore, we are removing this section from the revised manuscript.

Reviewer3: 5. *Cisplatin-induced apoptosis is not always perpetuates with drop of MMP. Release of cytochrome c and other intermembrane-located proteins might occur via formation of specific Bax/Bak-induced pores that lead to activation of caspase-9, etc, the process independent on Ca²⁺. Indeed, upon treatment with cisplatin caspase-9 is activating in cells expressing MICU1 (Fig. 3c) and removal of this protein is just accelerate this process. It is unclear why in cells with active caspase-9 PARP is not cleaved? It also raises the question concerning the role of calcium in this process. If removal of MICU1 just accelerates cisplatin-induced death, then the role of MICU1 in regulation of sensitivity to treatment is not the primary.*

Response: We appreciate the reviewer's thoughtful comments and regret mislabeling of the immuno blots. We would like to clarify that the band shown for PARP was actually representing the cleaved fragment of PARP (probed with cleaved PARP specific antibody, Cell signaling Technology CST#9541). However, to avoid the ambiguity we have now repeated the experiment and probed with PARP antibody (Cell signaling Technology CST #9532) to show both the total PARP and cleaved PARP fragment simultaneously (**Figure 3C**). Additionally we have performed the immunoblotting for cleaved caspase-3 and caspase-9 again. Our data suggests that cisplatin causes PARP cleavage in MICU1 silenced cells suggesting apoptosis induction, whereas the control cells did not show any PARP cleavage.

Reviewer3: 6. *The authors concluded that expression level of PDK and pPDH in ovarian cancers is higher than in normal cells. However, data from Fig 6a contradict it. Level of PDK in OSE cells is similar to OVCAR2 and OV90 cells. On the other hand, level of pPDH is not correlated with level of PDK. This discrepancy should be explained.*

Response: We appreciate the reviewer's thoughtful comments. Even though the PDK levels of OvCa cells OVCAR2 and OV90 are similar to OSE cells, importantly pPDH (ser293) levels in the ovarian cancer cells are higher indicating an aerobic glycolytic phenotype. MICU1 abrogates the uptake of Ca^{2+} into the mitochondria and thereby mitigates the activity of Ca^{2+} -dependent phosphatases (such as PDP) responsible for dephosphorylating and activating PDH. Therefore, although the levels of PDK might be similar in non-malignant and cancer cells, activity of PDP is rendered low in MICU1 expressing cancer cells resulting in pPDH accumulation. Furthermore, it is reported that the activity of glycolytic enzyme such as PDK is higher in cancer cells resulting in drug resistance and could be independent of the expression profile. [Sutendra et al., *Front Oncology* 3: 38 (2013)].

Reviewer3: 7. *Unfortunately, the total level of PDH in cells with siMICU1 is increasing (see Fig. 6C). In this case decreasing of the level of pPDH should be explained.*

Response: We appreciate the reviewer's concern and provided quantification of change in expression of pPDH upon MICU1 silencing. It is known that Pyruvate dehydrogenase kinase (PDK) phosphorylates PDH at ser293 that inhibits enzymatic activity of PDH, thereby, switches energy production towards aerobic glycolysis [Korotchkina et al., *The Journal of Biological Chemistry* 276, 5731-5738 (2001)]. Since MICU1 functions as a negative regulator of mitochondrial Ca^{2+} uptake, it inhibits the activity of Ca^{2+} -dependent phosphatases in the mitochondria such as PDP that dephosphorylates PDH and thus switches back energy requirement from glycolysis to pyruvic acid cycle. Therefore, MICU1 silencing in ovarian cancer cell will enhance Ca^{2+} overload in the mitochondria and thereby enhance activity of PDP resulting in dephosphorylation of PDH. Hence, we observe a reduction of pPDH upon MICU1 silencing. This cell line based observation is further supported in clinical cohort as a significant correlation between pPDH and MICU1 expression is observed with poor prognosis (**Figure 6L and Table S2**). The Spearman correlation coefficient between pPDH and MICU1 was significant among the various PFS groups ranging from 0.46 to 0.53. Furthermore, high pPDH expression is significantly correlated with low progression free survival (PFS), indicating aerobic glycolysis is associated with poor prognosis in OvCa patients (**Table S3**).

Reviewer3: 8. *Why shCTL-implantation group formed statistically significant larger tumors?*

Response: The shCTL group composed of OV90 cells when orthotopically implanted in bursa of mice ovary formed significant larger tumors because OV90 cells are reported to generate efficient orthotopic tumor models [Yakubov et al., *Neoplasia* 15(6): 609–619. (2013)]. In the absence of any therapeutic intervention tumor grows with time. We are demonstrating here that MICU1 silencing could potentially be an effective way to inhibit tumor growth and overcome therapy resistance.

Reviewer3: 9. *Data in Fig. 7a showed that treatment with cisplatin is equally efficient as removal of MICU1, which again raises the questions whether this removal sensitizes cells to treatment?*

Response: We appreciate the reviewer's thoughtful comments. We explained above related to the similar concerns raised by the reviewer 2. We discussed above that before initiation of the treatment animals were randomized in a blinded fashion. We observed that all the animals injected with shCTL cells developed large tumors (~1.469g), were moribund, appeared in distress and hence were euthanized along with the experimental groups on the same day to avoid undue suffering following the animal protocol approved by the institutional Animal Care and Use Committee (IACUC). We used same IACUC approved criteria to euthanize animals for all other groups in survival studies. We have now included the median survival values for the different groups (**Figure 7B**). It is evident from the table that cisplatin treatment in shCTL cells bearing animals yielded 2 extra days of survival while shMICU1 cells bearing animals survived for additional 13 days (43 days) and cisplatin therapy further prolonged their survival for additional 10 more days (53 days). It is also important to note that although the tumor growth inhibition by cisplatin seems to be proportionally identical between shCTL and shMICU1, in shMICU1 group 4 out of 5 animals showed negligible tumor formation upon cisplatin treatment (80% of the sample size) as compared to 1 out of 5 animals in shCTL+cisplatin group (20% of the sample size), supporting a role of MICU1 in drug resistance. However, we agree that there may be other pro-oncogenic functions of MICU1 that are currently unknown but will evolve over time.

Reviewer3: 10. *What happened with α -ketoglutarate- and isocitrate-dehydrogenases in cells deficient in MICU1?*

Response: We appreciate the reviewer's insightful comments and understand that other dehydrogenases like α -ketoglutarate (alpha-KG) and isocitrate-dehydrogenases (ISD) activity could also be affected by changes in $[Ca^{2+}]_m$ (Denton et al., 2009). However, in our model, MICU1 silencing had no effect either in the expression or activity of alpha-KG or ISD. The activity of these enzymes might require a particular threshold of Ca^{2+} for their activation, while PDH activity is enhanced in MICU1 silenced cells by activating Ca^{2+} -dependent pyruvate dehydrogenase phosphatases (PDP). We have included this observation of alpha-KD and ISD expression and activity in **Supporting Figure 17**.

Reviewer3: 11. *Discussion is too long and unclear.*

Response: We thank the reviewer for his/her suggestion to improve our manuscript and in accordance we have shortened the discussion section and provided more rationale information to establish and conclude our study.

REVIEWERS' COMMENTS:

Reviewer #1 (Remarks to the Author):

The authors have satisfactorily addressed the concerns raised in the first review. The paper is original and relevant in the field.

Reviewer #2 (Remarks to the Author):

I am satisfied with the authors reply to my comments, and I recommend the paper for publication.

Reviewer #3 (Remarks to the Author):

The authors addressed all my concerns and the revised version of the MS can be recommended for publication in Nature Communications